# AUDITING FAIRNESS ONLINE THROUGH INTERACTIVE REFINEMENT

## ABSTRACT

Machine learning algorithms are increasingly being deployed for high-stakes scenarios. A sizeable proportion of currently deployed models make their decisions in a black box manner. Such decision-making procedures are susceptible to intrinsic biases, which has led to a call for accountability in deployed decision systems. In this work, we focus on user-specified accountability of decision-making processes of black box systems. Previous work has formulated this problem as run time fairness monitoring over decision functions. However, formulating appropriate specifications for situation-appropriate fairness metrics is challenging. We construct AVOIR, an automated inference-based optimization system that improves bounds for and generalizes prior work across a wide range of fairness metrics. AVOIR offers an interactive and iterative process for exploring fairness violations aligned with governance and regulatory requirements. Our bounds improve over previous probabilistic guarantees for such fairness grammars in online settings. We also construct a novel visualization mechanism that can be used to investigate the context of reported fairness violations and guide users towards meaningful and compliant fairness specifications. We then conduct case studies with fairness metrics on three different datasets and demonstrate how the visualization and improved optimization can detect fairness violations more efficiently and ameliorate the issues with faulty fairness metric design.

## 1 INTRODUCTION

The use of advanced analytics and artificial intelligence (AI), along with its many benefits, poses important threats to individuals and broader society at large. Hirsch et al. (2020) identify: invasion of privacy; manipulation of vulnerabilities; bias against protected classes; increased power imbalances; error; opacity and procedural unfairness; displacement of labor; pressure to conform, and intentional and harmful use as some of the key areas of concern. A core part of the solution to mitigate such risks is the need to make organizations accountable and ensure that the data they leverage and the models they build and use are both inclusive of marginalized groups and resilient against societal bias. Deployed AI and analytic systems are complex multi-step processes that can produce several sources of risk at each step. At each of these stages, determining accountability in the decision making in AI processes requires a determination of who is accountable, for what, to whom, and under what circumstances (Nissenbaum, 1996; Cooper et al., 2022). A more comprehensive overview of the mechanisms that can support accountability with respect to the different stages of design of a machine learning system ca be found in the work of Cooper et al. (2022). We center our analysis on the sub problem of auditing barriers towards investigating claims surrounding mathematical guarantees of automated decision making processes. Governments across the world are wrestling with the implementation of auditing regulation and practices for increasing the accountability of decision processes. Recent examples include the New York City auditing requirements for AI hiring tools (Vanderford, 2022), European data regulation (GDPR 2018), accountability bills 2019; 2021 and judicial reports 2018. These societal forces have led to the emergence of checklists (Mitchell et al., 2019; Sokol & Flach, 2020), metrics of fairness (Verma & Rubin, 2018), and recently, algorithms and systems that observe and audits the behavior of AI algorithms. Such ideas date back to the 1950s (Moore, 1956) but research has largely been sporadic until very recently with the widespread use of AI-based decision making giving rise to the vision of algorithmic auditing (Galdon Clavell et al., 2020). We present a framework for *Auditing and Verifying fairness Online through Interactive*

> Contextualizing wrt Nissenbaum

Figure 1: Shaded nodes describe our contributions in the AVOIR framework.

*Refinement* (AVOIR) [1]. AVOIR builds upon the ideas on distributional probabilistic fairness guarantees (Albarghouthi & Vinitsky, 2019; Bastani et al., 2019), generalizing them to real-world data. An overview of AVOIR is provided in Figure 1.

## 1.1 PRELIMINARIES

Machine learning testing (Zhang et al., 2020) is an avenue that can be used to expose undesired behavior and improve the trustworthiness of machine learning systems. Fairness criteria quantify the relationship between the outcome metric across multiple subgroups or similar individuals among the population. Formal definitions of fairness focus on observational criteria, i.e., those that can be written down as a probability statement involving the joint distribution of the features, sensitive attributes, decision making function, and actual outcome. Our framework, AVOIR, supports implementing a large range of group fairness criteria, including demographic parity (Calders et al., 2009), equal opportunity (Hardt et al., 2016), disparate mistreatment (Zafar et al., 2017), and various combinations of these criteria. As an example, suppose $r \in \{0, 1\}$ denotes the return value of a binary decision function (say, candidate selection for a job), and $s$ is an indicator denoting whether a candidate belongs to a minority population. The 80%-rule for disparate impact (EEOC, 1979; Feldman et al., 2015)is a fairness criterion which states that

$$\frac{\Pr[r = 1|s]}{\Pr[r = 1|\neg s]} \geq 0.8$$

When implemented in the AVOIR DSL grammar, the above 80%-rule would be the specification E[r|S==s] / E[r|S!= s] >= 0.8. Here, the term E[r|S!=s]/E[r|S == s] is a *subexpression* of the specification. The smallest units involving an expectation (eg., E[r|S!=s]) are denoted as an *elementary subexpressions*. Our algorithm works by using adaptive concentration sets (Zhao et al., 2016; Howard et al., 2021) to build estimates for *elementary subexpressions*, and then deriving the estimates for expressions that combine them. We aim to derive statistical guarantees about fairness criteria based on estimates from observed outputs. For example, let $X$ be an observed Bernoulli r.v[2], then an assertion $\phi_X = (\overline{\mathbb{E}}[X], \varepsilon, \delta)$ over $X$, corresponds to an estimate satisfying

$$\phi_X \equiv \Pr[|\mathbb{E}[X] - \overline{\mathbb{E}}[X]| \geq \varepsilon] \leq \delta \tag{1}$$

where $\overline{\mathbb{E}}[X]$ denotes an empirical estimate of $E[X]$. We then use assertions $\phi_X, \phi_Y$ to assert claims for expressions involving $X, Y$. For example, for the 80%-rule, assertions over $X/Y$. A specification involves either a comparison of expressions with constants (eg., $X/Y > 0.8$), or a combination of multiple such comparisons. Such a specification may be True ($T$) or False ($F$) with some probability. For a given specification $\psi$, we denote the claim that $P[\psi = F] \geq 1 - \delta$ as $\psi : (F, \delta)$, where $\delta$ denotes the failure probability of a guarantee. Given a stream of (observations, outcomes from the decision functions), and a specified threshold probability $\delta$, we will continue to refine the estimate for a given specification until we reach the threshold. We focus on fairness criteria that can be expressed using Bernoulli r.v. as it allows the simplification of probabilities into expectation, as $\Pr[r = 1] = \mathbb{E}[r]$. Specifications involving variables that take more than two values can be implemented using transformations and boolean operators (examples in Appendix H).

## 1.2 RELATED WORK

There are a plethora of fairness criteria and subtle changes in their definition can change the implications on decision making (Castelnovo et al., 2021). Practitioners need support when selecting, designing, and guaranteeing fairness for deployed machine learning algorithms. Prior work on fairness has helped develop nuanced notions and algorithms to help train more 'fair' machine learning models. These include group fairness measures such as, inter alia, minimizing disparate impact (Calders et al., 2009; Feldman et al., 2015), maximizing the equality of opportunity (Hardt et al., 2016) In contrast with group fairness notions, causal notions of fairness Kusner et al. (2017) and individualized notions of fairness Dwork et al. (2012) provide alternative statistical mechanisms for understanding discriminatory behaviors of automated decision systems. Thomas et al. (2019) proposed the Seldonian Framework as a generic mechanism for model users to design algorithms that help train machine learning models that can regulate them against undesirable behaviors.

We focus on the problem of detecting and diagnosing whether systems designed under any framework follow any prescribed regulatory constraints that are supported within the grammar of AVOIR. That is, we are agnostic to the design itself; rather, we are interested in testing the adherence of models to specified criteria. We use a probabilistic framework to verify this behavior. Alternative frameworks such as the AI Fairness 360 (Bellamy et al., 2019) provide mechanisms to quantify fairness uncertainty, though they are restricted to pre-supported metrics. Uncertainty quantification (Ghosh et al., 2021b; Ginart et al., 2022) is an alternative mechanism to provides adaptive guarantees, however, existing work is designed for commonly used outcome metrics such as accuracy, F1-score, etc., rather than for fairness metrics. Justicia (Ghosh et al., 2021a) optimizes uncertainty for fairness metrics estimates using stochastic SAT solvers but can only be applied to a limited class of tree-based classification algorithms.

> More nuanced comparison

Prior work on fairness testing is most closely related to AVOIR. Fairness testing (Galhotra et al., 2017) provides a notion of causal fairness and generates tests to check the fairness of a given decision-making procedure. Given a specific definition of fairness, Fairtest (Tramèr et al., 2017) and Verifair (Bastani et al., 2019) build a comprehensive framework for investigating fairness in data-driven pipelines. Fairness-aware Programming (FP) (Albarghouthi & Vinitsky, 2019) combined the two demands of machine learning testing and fairness auditing to make fairness a first-class concern in programming. Fairness-aware programming applies a runtime monitoring system for a decision-making procedure with respect to an initially stated fairness specification. The overall failure probability of an assertion is computed as the sum of failure probabilities of each constituting sub-expression (using the union bound). FP does not provide any specific mechanism for splitting

---

[1]AVOIR in French means "to have" and this acronym reflects both our aspirational goal to achieve fairness in advanced analytics and AI but also reflects what is currently verifiable given a dataset, a model and a fairness specification.

[2]random variable

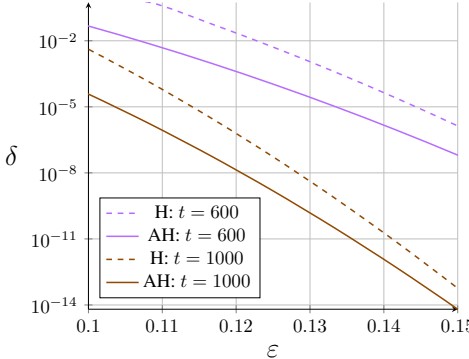

(a) At the same concentration $\varepsilon$, lower failure probability $\delta$ for the majority class.

$$\langle spec \rangle ::= \langle ETerm \rangle \langle comp\text{-}op \rangle \text{ c}$$
$$\mid \quad \langle spec \rangle \wedge \langle spec \rangle$$
$$\mid \quad \langle spec \rangle \vee \langle spec \rangle$$

$$\langle ETerm \rangle ::= \mathbb{E}[\langle E \rangle]$$
$$\mid \quad \mathbb{E}[\langle E \rangle | \langle E \rangle]$$
$$\mid \quad \text{c} \in \mathbb{R}$$
$$\mid \quad \langle ETerm \rangle \ \{+, -, \times, \div\} \ \langle ETerm \rangle$$

(b) $\langle E \rangle$ refers to pure expressions and $\langle comp - op \rangle$ is any comparison operator $\in \{>, <, =, \neq\}$.

Figure 2: *(Left)* Failure probability of Bernoulli r.v. being concentrated around its mean for different $n$. H = (online) Hoeffding, AH = Adaptive Hoeffding. *(Right)* Modified Grammar for Specification.

uncertainty, and Verifair splits it equally across all constituent *elementary subexpressions*. Thus, assertion bounds for subexpressions in both FP and Verifair are split inefficiently. Proving guarantees for overall uncertainty across multiple groups can be ameliorated by balancing it across subexpressions with differences in the number of observed samples. For example, consider Bernoulli r.vs $X_{1,2}$ for which we derive concentration guarantees $\Pr[|\mathbb{E}[X_i] - \overline{\mathbb{E}}[X_i]| \geq \varepsilon_i] \leq \delta_i$ after $t_i$ observations. From the Hoeffding inequality, $\delta = 2e^{-2t\varepsilon^2}$. We can claim tighter guarantees for $X_2$ if $t_2 > t_1$ as the failure probability is lower at the same concentration $\varepsilon$. That is, $\varepsilon_1 = \varepsilon_2, t_2 > t_1 \implies \delta_1 > \delta_2$. Varying $\varepsilon$ across subexpressions to minimize the overall $\delta = \delta_1 + \delta_2$ allows convergence in fewer iterations. This observation motivates us to optimize over sub-expressions and provide tighter overall concentration for compound expressions. Adaptive versions of these inequalities also have similar patterns (see Figure 2a).

> Clarify the concentration.

### 1.3 AVOIR: KEY CONTRIBUTIONS

We now summarize our contributions vis-à-vis FP and Verifair. (1) We build up AVOIR in the framework of **confidence sets** (Howard et al., 2021) which enables the **adaptive optimization** of $\delta$ across subexpressions. Note that FP provides examples with equal splits across two terms though it makes no specific prescription of splits. Verifair splits uncertainty equally across all elementary subexpressions. (2) The confidence sets framework allows us to move away from assuming a known data distribution or alternatively, fitting a density estimator over the population prior to fairness testing, required in Verifair. (3) We augment the **bound propagation rules** to facilitate the online optimization process and allow propagation of constraints along with assertions at each iteration. (4) We build an **inference engine** that supports automated inference of propagation rules for wide range of metrics. In Section 3, we provide examples of inference over specifications involving over two subexpressions, which are not possible without extending the implementations provided by previous work. As a baseline, we also implement bound inference rules from Verifair (denoted AVOIR-VF). (5) We support **interactive diagnosis** of fairness specification violations using visual cues associated with convergence of subexpressions. We demonstrate the use of these cues to help drive the design of specifications in Section 3.2, which show how a user may have audited their original claim and refined mathematical bounds.

> New contributions section.

## 2 AVOIR FRAMEWORK

### 2.1 LANGUAGE SPECIFICATION

We describe AVOIR's Domain Specific Language (DSL) used for specifying fairness metrics. Concrete examples of implemented specifications in AVOIR's DSL are provided in Section 3. We focus on binary decision making functions; their outputs can be characterized by Bernoulli r.v.s. Note that for such a Bernoulli r.v. $X$, $\mathbb{E}[X] = \Pr[X = 1]$ and hereafter, these are used interchangeably. Fairness specifications are implemented as decorators over decision functions. Consider a decision function $f : X \to \{0, 1\}$, where $X = (X_1, \ldots, X_k)$ denotes a real-valued input vector. We use $R = f(X)$ to simplify the remainder of the definitions.

- To support expressions beyond those that produce binary outputs, we use the grammar to construct Bernoulli r.vs. For example, a $\nu$-threshold based real-valued output, $R' = (R > \nu)$ and a multi-class output, for class $j$, $R' = (R == j)$ correspond to Bernoulli r.vs.

- Expressions involving $R$ and $X_i$ act as the arguments <E> to construct an <ETerm>. For example $\mathbb{E}[R > 0 | X_1 + X_2 > a]$ is an *elementary subexpression* and an <ETerm>

> Expanded definitions

$c$ terms represent constant real values, used, for example, as bounds for expressions. The grammar provided in Figure 2b can be then used to construct various group fairness criteria. We modified the grammar from prior work to include two additional operations. First, we added a `given` argument to the expectation term, which allows a user to specify conditional probabilities directly, in contrast to specifying it as a ratio of joint/marginal probabilities.

$$\frac{\mathbb{E}(A \vee (B = b))}{\mathbb{E}(B = b)} \to \mathbb{E}(A, \texttt{given} = (B = b))$$

which is used to represent $\mathbb{E}[A|B = b]$, simplifying expressions used for group fairness specification. Additionally, we add binary comparison operators $<, >, ==, ! =$, which further simplifies the process of writing specifications.

## 2.2 PROPAGATING BOUNDS

Generating the bounds for a specification requires propagating guarantees from elementary subexpressions. Assuming that observed values for each `<E>` correspond to an underlying random variable $X$, a probabilistic guarantee $\phi_X$ consists of an empirical estimate $\overline{\mathbb{E}}[X]$, a concentration bound $\varepsilon_X$, and a failure probability $\delta_X$, such that $\Pr[|\mathbb{E}[X] - \overline{\mathbb{E}}[X]| \geq \varepsilon_X] \leq \delta_X$. We refer to expressions of this form as *elementary* subexpressions. A fairness specification will typically consist of multiple such elementary expressions, denoted as *compound* expressions. For compound expressions, we must infer the implied guarantees that can be provided, with corresponding constraints. Each inference rule corresponds to a derivation in the DSL grammar. Inference rules have preconditions and postconditions that follow the general expression

$$\frac{\bigcup \{r | r \in \{\phi, \psi, C\}\}}{\bigcup \{s | s \in \{\phi, \psi, C\}\}}$$

where $\phi$ denotes a claim for a subexpression, $\psi$ for a `<spec>`, $\overline{\mathbb{E}}$ and $\varepsilon$ are the mean and concentration terms associated with a subexpression claim, $C$ denotes a constraint. For example, consider starting with the assumptions $X : (\overline{\mathbb{E}}[X], \varepsilon_X, \delta_X), Y : (\overline{\mathbb{E}}[Y], \varepsilon_Y, \delta_Y)$. Then we have

$$|\mathbb{E}[X] \pm \mathbb{E}[Y] - (\overline{\mathbb{E}}[X] \pm \overline{\mathbb{E}}[Y])| = |(\mathbb{E}[X] - \overline{\mathbb{E}}[X]) \pm (\mathbb{E}[Y] - \overline{\mathbb{E}}[Y])|$$
$$\leq |\mathbb{E}[X] - \overline{\mathbb{E}}[X]| + |\mathbb{E}[Y] - \overline{\mathbb{E}}[Y]|$$
$$\leq \varepsilon_X + \varepsilon_Y$$

Updated the language to show the assumptions

i.e., we can derive $X \pm Y : (\overline{\mathbb{E}}[X] \pm \overline{\mathbb{E}}[Y], \varepsilon_X + \varepsilon_Y, \delta_X + \delta_Y)$. Some derivations also lead to rules that require constraints. For instance, assume $X : (\overline{\mathbb{E}}[X], \varepsilon_X, \delta_X), \overline{\mathbb{E}}[X] > c$. Then we have $\Pr[X < \overline{\mathbb{E}}[X] - \varepsilon_X] > 1 - \delta$ If we add the constraint that $\overline{\mathbb{E}}[X] - \varepsilon_X \geq c$, we have $\Pr[X < c] > 1 - \delta$, thus,

$$X : (\overline{\mathbb{E}}[X], \varepsilon_X, \delta_X) \implies \psi \equiv X > c : (T, \delta_X)$$
$$\text{under the constraint } \{\overline{\mathbb{E}}[X] - \varepsilon_X \geq c\}$$

The full set of inference rules required for the DSL is provided in the appendix (Figure 5). The implementation in AVOIR follows these rules but can be extended to other rule inference templates that support the DSL. We note that these rules extend the ones implemented by VeriFair (VF)[3] (Bastani et al., 2019) with constraints that enable the optimizations required in AVOIR (see Appendix A).

## 2.3 OPTIMIZING BOUNDS

### 2.3.1 AVOIR ALGORITHM

The pseudocode for the optimization procedure in AVOIR is described in the appendix (Algorithm 1). The input to the algorithm is the reporting threshold probability $\Delta$ and a specification $\psi$. We then infer a symbolic optimization problem is inferred corresponding to the failure probabilities and constraints derived from concentration bounds. At each step, the `OBSERVE(X)` function is called with new observation of every *elementary* subexpression and observed output. The running mean and counts of observations are updated. The final optimization problem `OPT` corresponding to each specification is a nonlinear constrained optimization problem. We use the COIN-OR implementation of IPOPT (Wächter & Biegler, 2006), accessed though the Pyomo (Hart et al., 2011) interface to solve this problem at each step. If a solution is successfully found for `OPT`, the algorithm terminates, with the estimate for the specification having reached the required threshold. If no solution is found, the estimates continue to be updated with $\delta_i = \Delta$ for each *elementary* subexpression. The main intuition behind the algorithm is to create a confidence sequence corresponding to

---

[3]Verifair

the estimates at each time step. The `OPT` corresponding to a specification:

$$
\min_{\delta_i} \sum_{i=1}^{n} \delta_i
$$
$$
\text{s.t. } g_k(\delta_{1,\ldots,n}, \overline{\mathbb{E}}[X_1], \ldots, \overline{\mathbb{E}}[X_n]) \leq \varepsilon_k \tag{2}
$$
$$
0 \leq \delta_i \leq 1
$$

where $g_k$ and $\varepsilon_k$ are the functions/bounds derived using the transformations carried out through the DSL inference rules (further details in Appendix A.2).

**Definition 1.** *For $\delta \in (0, 1)$, a $(1 - \delta)$ confidence sequence is a sequence of confidence sets, usually intervals $(\mathrm{CI_t})_{t=1}^{\infty}$,, say $\mathrm{CI_t} := (\mathrm{L_t}, \mathrm{R_t}) \subseteq \mathbb{R}$ satisfying a uniform convergence guarantee. After observing the tth unit, we calculate an updated confidence set $\mathrm{CI_t}$ for an unknown quantity of interest $\theta_t$ with the coverage property $\Pr(\forall t \geq 1, \theta_t : \theta_t \in \mathrm{CI_t}) \geq 1 - \delta$ (Howard et al., 2021).*

In this paper, we focus on the mean of r.v.s $\mathbb{E}[X]$ that constitute estimates for *elementary* subexpressions as the quantities of interest. We use adaptive concentration inequalities to construct these confidence sequences. Any adaptive concentration inequality that can be applied to a r.v. $X \in \{0, 1\}$ such that

$$
\Pr[|\overline{\mathbb{E}}_t[X] - \mathbb{E}[X]| \geq \varepsilon(t, \delta)] \leq \delta \tag{3}
$$

can be used in AVOIR. Here, $\overline{\mathbb{E}}_t[X]$ denotes the empirical estimate of $\mathbb{E}[X]$ after the $t^{\text{th}}$ observation. For the purpose of comparison with previous work (eg., VF), we use the Adaptive Hoeffding Inequality (Zhao et al., 2016), which will be referred to as AIN hereafter.

**Theorem 1.** *The sequence of estimates generated by AVOIR form a confidence set.*

The proof follows from the fact that AVOIR always estimates using a failure probability higher than that which is provided by AIN, and hence applying a union bound ensures that the estimates are a confidence set. The full proof is provided in Appendix C.

**Corollary 1.1.** *The estimates for the overall specification $\psi$ form a confidence sequence converging to $\psi : (b, \Delta), b \in \{T, F\}$.*

*Proof.* We initialize the main specification with the required failure probability $\Delta$. The termination condition requires $\sum \delta_i \leq \Delta$. From Theorem 1 we can infer that the confidence sequence corresponding to the termination achieves the required threshold $\Delta$, and therefore, is valid. $\square$

### 2.3.2 IMPROVEMENTS OVER BASELINE

In all prior work (Albarghouthi et al., 2017; Albarghouthi & Vinitsky, 2019; Bastani et al., 2019), $\delta_i$ for each *elementary* subexpressions is set to $\Delta/n$, where $n$ is the number such term in the specification. This simplification is carried out using the assumption $A_\delta := \delta_i = \delta_j \forall i, j$ for all *elementary* subexpressions. As we do not make this assumption, we can prove the following key theorem.

Introduce $A_\delta$

**Definition 2.** *We define the specification stopping time $\mathcal{T}$ for a confidence sequence as the smallest time t such that, given a threshold $\Delta$ and a specification $\psi$, we can terminate any inference algorithm to claim that $\Pr[\forall t \geq 1, \psi_t = \widehat{\psi}_{\mathcal{T}}] \geq 1 - \Delta$, where $\widehat{\psi}_{\mathcal{T}}$ is the estimate of $\psi$ at time $\mathcal{T}$.*

**Theorem 2.** *Given a threshold probability $\Delta$ for a specification $\psi$, let the stopping time for AVOIR be $\mathcal{T}$ and stopping time with the $A_\delta$ assumption be $\mathcal{T}^+$. Then $\mathcal{T} \leq \mathcal{T}^+$*

See Appendix D for the proof.

**Concrete Example**  Consider a Bernoulli r.v $R$ corresponding to the output of a binary decision function, with $s$ being an indicator of class membership. Let $X = r \vee s$ and $Y = r \vee \neg s$ be r.vs corresponding to a positive decision for the majority and minority classes, respectively. Suppose we aim to estimate $\psi := X - Y < \varepsilon_T$

We demonstrate the improvements possible using our approach by instantiating this example with data. Suppose we want the upper bound of the failure probability $\Delta = 0.1$ for the specification. Consider a set of observations such that $\overline{\mathbb{E}}[X] = 0.8, n_X = 1550$ and $\overline{\mathbb{E}}[Y] = 0.5, n_Y = 310$.

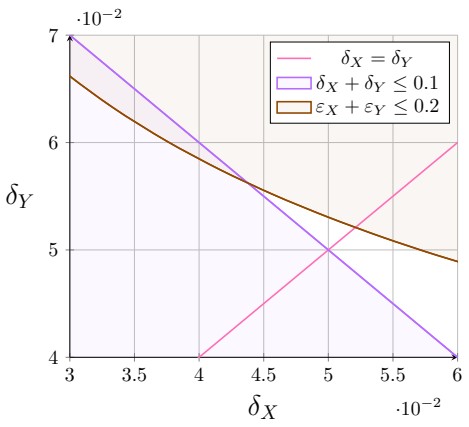
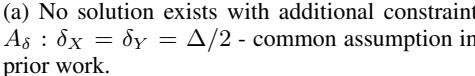

(a) No solution exists with additional constraint $A_\delta : \delta_X = \delta_Y = \Delta/2$ - common assumption in prior work.

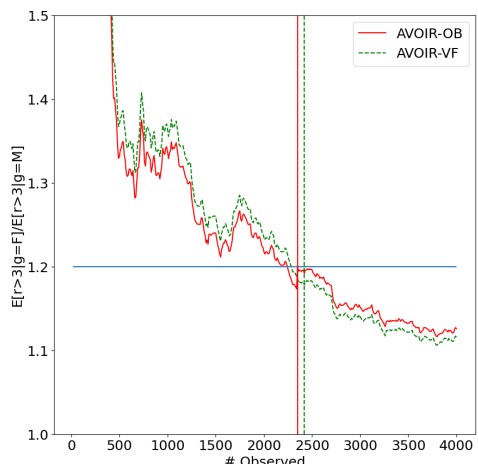

(b) Bounds for first half of a gender-fairness specification generated by AVOIR-OB and AVOIR-VF.

Figure 3: *(Left)* AVOIR finds a solution for a *theoretical* scenario with $\delta_X + \delta_Y \leq \Delta$ under constraint $\varepsilon_X + \varepsilon_Y \leq \varepsilon_T$ *(Right)* For *RateMyProfs*, a real-world dataset, the vertical lines show the step at which the methods can provide a guarantee of failure for the upper bounds with $\Delta <= 0.05$.

Figure 3a shows that no solution is feasible for the optimization problem with $A_\delta$. However, AVOIR can find a solution. For the optimal solution, $\delta_2 \approx 2.35\delta_1$, which aligns with our intuition from section 1.2 about allocating higher failure probability to terms with the majority of observations. The optimization problem inferred by AVOIR:

$$
\min_{\delta_X, \delta_Y} \delta_X + \delta_Y
$$
$$
\text{s.t. } \varepsilon_X + \varepsilon_Y \leq \overline{\mathbb{E}}[X] - \overline{\mathbb{E}}[Y] - \varepsilon_T \tag{4}
$$
$$
0 \leq \delta_{X,Y} \leq 1
$$

## 2.4 VISUALIZATION FOR INTERACTIVE REFINEMENT

Using our specification framework as a backend, we built an interactive application for analysis and refinement of specifications provided in our grammar. Specifically, fairness specifications can be naturally parsed into a tree because of the structure of the grammar. Each node of the tree represents some sub-expression in the syntax tree of the overall specification. These nodes allow a user of AVOIR to interactively audit and tune the specification definition. To create the visualization, we use Vega (Satyanarayan et al., 2015), a declarative JSON-based visualization grammar. We log the estimates during runs of AVOIR and then output the grammar in a tabular JSON-format that contains a row for each grammar element and its associated evaluations. This tabular data is used by our Vega specification to produce the visualizations. By selecting one of the nodes in the syntax tree, a user can see a plot of the evaluation values associated with the selected grammar element. This allows for comparison of multiple grammar elements. The ability to analyze and compare these evaluation values provides context surrounding specification violations, and assists the user in interacting with and deciding how to refine a specification We provide a detailed example of how these interactions can help AVOIR users choose an appropriate fairness metric in Section 3.

## 3 CASE STUDIES

The following text describes two real-world scenarios for AVOIR. We implement Verifair inference rules in the AVOIR (denoted as AVOIR-VF) framework, which allows us to sidestep the assumptions of a having a known data generating distribution, making it more efficient than Verifair. We denote AVOIR-OB as the implementation which also utilizes the constraints and optimization framework. An important case study on the COMPAS dataset can be found in Appendix G.

Explaining
AVOIR-VF

## 3.1 RATE MY PROFS

In this section, we provide a detailed black-box machine learning model (ML) based case study on a real-world dataset. In this case study, we use the rate my professors (RMP) dataset released by Keymanesh et al. (2021). This dataset includes professor names and reviews for them written by students in their classes, ratings, and certain self-reported attributes of the reviewer. Ratings are provided on a five-point scale (1-5 stars). We use the preprocessing described in Keymanesh et al. (2021) to infer the gender attribute for the professors. This dataset is divided into an 80-20 split (train-test). We then train a BERT-based transformer model Devlin et al. (2019) on the training split. We use the implementation from the simpletransformers[4] package. The loss function chosen is the mean-squared error from the true ratings. On the test set, we track a gender-fairness specification in the model outputs:

(E[r $>$ 3 | gender = F] / E[r $>$ 3 | gender = M $<$ 1.2) &
(E[r $>$ 3 | gender = M)] / E[r $>$ 3 | gender = F] $>$ 0.8)

We set the failure probability $\Delta = 0.05$. OPT is run after each batch (5 items/batch). Figure 3b shows that AVOIR-OB[5] can provide a guarantee in **2.5**% fewer iterations than AVOIR-VF. Note also that the OB guarantee provided tries to optimize for the failure probability while staying under the required threshold, remaining closer to the required threshold in subsequent steps.

## 3.2 ADULT INCOME

In this case study we use the Adult income dataset (Kohavi, 1996) which has been used frequently in prior fairness-related work. The historical dataset labels individuals from the 1994 census as having a *high-income* ($> 50,000$ a year) or not ($\leq 50,000$ a year). In this case study, we look at a column of data as a black-box measurement (internally, we use a materialized view, details in Appendix G.1).

US Federal laws mandate against race and sex based discrimination. Thus, the specification we start our analysis with is a group fairness property that monitors the difference of the proportions of individuals with sex recorded as male that have a high income to females that have a high income should be less than $0.5$. In addition, we ensure that the difference between individuals with race marked as white and those without should have a difference of less than $0.5$. The associated specification is given below, where h is an indicator for whether an individual is *high-income* is the binary classification output of our model:

(E[h | sex=M] $-$ E[h | sex=F] $<$ 0.5) & \
(E[h | race=W] $-$ E[h | race !=W] $<$ 0.5)

In this example, we set the failure threshold probability $\Delta = 0.15$

When run with this specification, the generated materialized view cannot achieve the required bound. We can then use our iterative refinement visualization tool to analyze different components of the specification. A developer would first interact with the left subtree of the specification. Due to paucity of space, this visualization is presented in Appendix G.2. The plot for the corresponding data is shown in Figure 4a shows that guarantees cannot converge under the threshold with the given number of data samples. The developer can now choose to either reduce the guarantee (i.e. reduce $\delta$) or increase the threshold. Next, analyzing the right subtree, the race group fairness term can be guaranteed to be under the threshold (Figure 4b). Using this information, the developer can make an intelligent decision to increase the threshold on the group fairness for term for sex. Suppose they increase it to $0.55$ and rerun the analysis. OB is able provide a guarantee at this threshold within 870 steps, whereas VF can provide it at 960 steps, demonstrating a relative improvement of about **10.35**%. Additionally, the optimal $\delta$ split across the terms are $\approx (0.135, 0.36 * 10^4)$ which is far from the equal split allocated by VF. The reason for this split is because increasing the threshold for the first time provides the optimizer with additional legroom to better distribute the failure probabilities between the two terms.

---

[4]https://simpletransformers.ai/
[5]OB = Optimized Bounds

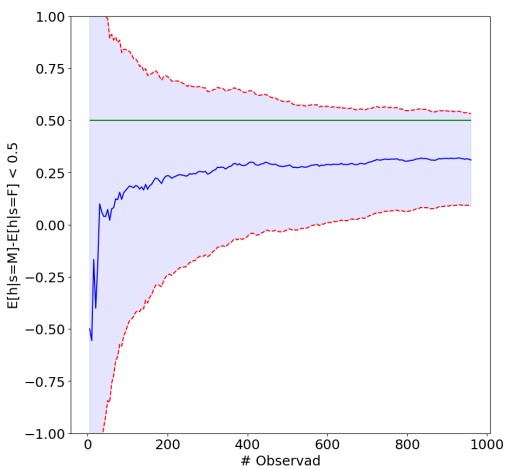 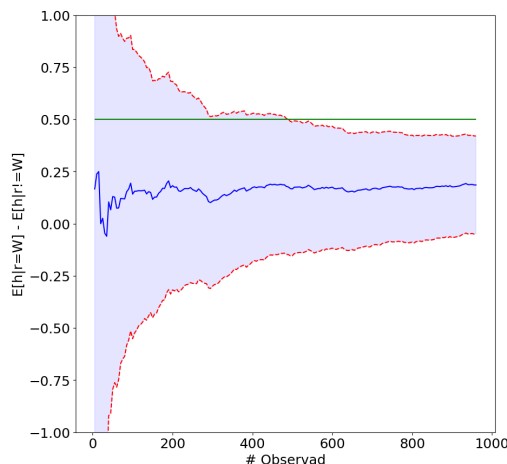

(a) Group fairness for sex. Difference in ratio of high income earners in left subtree for initial specification.

(b) Group fairness for race (difference in ratio of high income earners) in right subtree for initial specification.

Figure 4: *(Left)* Red dotted lines the upper bound of the value cannot be guaranteed to be under the threshold at the specified failure probability. *(Right)* Guarantee possible with given data.

## 4 DISCUSSION & FUTURE WORK

The case studies presented in the previous section demonstrate the ability of our tools to provide vital context when deciding how to refine a model or fairness specification. Although this contextual information makes decisions easier, it is not always clear how one should alter a specification in light of a violation and its relevant context. To assist in these decisions, we are currently examining ways work to suggest edits that are likely to achieve the desired intent of a developer. Using our visual analysis tool for refinement, we can gather edits from developers and then use that data to learn iterative changes to the syntax tree of the specification. In addition to improving the usability of our tools for making fairness specification refinements, we also envision a more scalable framework. Our case studies look at a single model with respect to a single dataset. However, real-world deployment of machine learning often contain many clients with models and datasets that may evolve and drift over time. We take it as future work to study the efficient monitoring of machine learning behavior with respect to a fairness specification in a distributed context, enabling horizontal scalability. We believe techniques such as decoupling the observation of data and the reporting results from the monitoring of the results are promising and can lead to the desired scalability.

Non stationary data is future work

## 5 CONCLUSION

We present the AVOIR framework for easily defining and monitoring fairness specifications online and aids in the interactive refinement of specifications. AVOIR is easy to integrate within modern database systems but can also serve as a standalone system evaluating whether black box machine learning models are meeting specific fairness criteria on specific datasets (including both structured and unstructured data) as described in our case studies. AVOIR extends the grammar from Fairness Aware Programming Albarghouthi & Vinitsky (2019) with operations that enhance expressiveness. In addition we derive probabilistic guarantees that improve the confidence with which specification violations are reported. To assist in refinement of specifications, we develop an interactive visual analysis application within AVOIR. Through case studies, we demonstrate that AVOIR can provide users with insights that contribute directly to refinement decisions. Our framework builds the foundation for further improvements to the fairness specification, auditing and verification workflow. We plan to extend this work to provide intelligent specification refinement suggestions and support distributed machine learning settings. We also plan to explore the use of AVOIR for fair ranking problems and tailored database integration.

## 6 REPRODUCIBILITY

To enhance the reproducibility of our work, on the theoretical side, all proofs (with the necessary assumptions) are provided in the appendix. Specifically, proofs for the inference engine are in Appendix A, and proofs for the correctness of bounds are provided in Appendix C. Theorem 2, which shows how AVOIR improves over prior work is proved in Appendix D. To reproduce the results of the case studies in the paper, each case study is encapsulated inside a Jupyter notebook. These notebooks are attached along with the source code for AVOIR. In addition, all datasets used for generating results for the case studies are also attached in the submitted supplementary documentation. Finally, the model weights used for the *RateMyProfs* study for exact reproduction are provided in a dropbox folder hosted at `https://www.dropbox.com/sh/n5o4vswnkxv34zr/AABthgLMaYL3MuA0KC39Z1G8a?dl=0`.

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

## A INFERENCE RULES

In Figure 5, we provide a set of rules that can be used to determining the constraints and guarantees associated with a specification. We represent

$$X \odot Y : (E, \varepsilon, \delta) \equiv \Pr\left(|\mathbb{E}[X] \odot \mathbb{E}[Y] - E| \geq \varepsilon\right) \leq \delta$$

where $\odot$ represents a binary operator. Constraints are represented in curly brackets $\{\}$.

The proof of correctness for each inference rule starts from the assumptions above the horizontal line and derives the assertions below. These proofs use ideas similar to those in Bastani et al. (2019). We reproduce the proofs in Appendix A.1 here for completeness. Note that the assertions in the base case (elementary subexpressions) can be arrived at by applying the Adaptive Hoeffding INequality, (AIN).

AIN

### A.1 INFERENCE RULES WITH CONSTRAINTS

In Section 2.2 we provided the proofs for $X \pm Y$, $X > c$. In the following text we provide the proofs for the remainder of the inference rules.

**Product** Starting with $\phi_X = X : (\overline{\mathbb{E}}[X], \varepsilon_X, \delta_X)$, $\phi_Y = Y : (\overline{\mathbb{E}}[Y], \varepsilon_Y, \delta_Y)$. First, from union bound, both of these hold true with probability at least $1 - \delta_X - \delta_Y$. Then,

$$
\begin{aligned}
|\mathbb{E}[X]| &= |\overline{\mathbb{E}}[X] - \overline{\mathbb{E}}[X] + \mathbb{E}[X]| \\
&\leq ||\overline{\mathbb{E}}[X] + |\overline{\mathbb{E}}[X] + \mathbb{E}[X]| \\
&\leq ||\overline{\mathbb{E}}[X]| + \varepsilon_X
\end{aligned}
$$

We have

$$
\begin{aligned}
|\overline{\mathbb{E}}[X]\overline{\mathbb{E}}[Y] - \mathbb{E}[XY]| &= |\overline{\mathbb{E}}[X]\overline{\mathbb{E}}[Y] - \mathbb{E}[X]\mathbb{E}[Y]| && \text{(as } X, Y \text{ Bernoulli)} \\
&= |\overline{\mathbb{E}}[X]\overline{\mathbb{E}}[Y] - \overline{\mathbb{E}}[X]\mathbb{E}[Y] + \overline{\mathbb{E}}[X]\mathbb{E}[Y] - \mathbb{E}[X]\mathbb{E}[Y]| \\
&= |\overline{\mathbb{E}}[X](\overline{\mathbb{E}}[Y] - \mathbb{E}[Y]) + \mathbb{E}[Y](\overline{\mathbb{E}}[X] - \mathbb{E}[X])| \\
&\leq |\overline{\mathbb{E}}[X]||(\overline{\mathbb{E}}[Y] - \mathbb{E}[Y])| + |\mathbb{E}[Y]||(\overline{\mathbb{E}}[X] - \mathbb{E}[X])| \\
&\leq |\overline{\mathbb{E}}[X]|\varepsilon_Y + |\mathbb{E}[Y]|\varepsilon_X \\
&\leq |\overline{\mathbb{E}}[X]|\varepsilon_Y + (|\overline{\mathbb{E}}[Y]| + \varepsilon_Y)\varepsilon_X \\
&= |\overline{\mathbb{E}}[X]|\varepsilon_Y + |\overline{\mathbb{E}}[Y]|\varepsilon_X + \varepsilon_X\varepsilon_Y
\end{aligned}
$$

Therefore, $X \times Y : (\overline{\mathbb{E}}[X]\overline{\mathbb{E}}[Y], \varepsilon_X\varepsilon_Y + \overline{\mathbb{E}}[X]\varepsilon_Y + \overline{\mathbb{E}}[Y]\varepsilon_X, \delta_X + \delta_Y)$

$$\frac{X : \left(\overline{\mathbb{E}}[X], \varepsilon_X, \delta_X\right), Y : \left(\overline{\mathbb{E}}[Y], \varepsilon_Y, \delta_Y\right)}{X \pm Y : \left(\overline{\mathbb{E}}[X] \pm \overline{\mathbb{E}}[Y], \varepsilon_X + \varepsilon_Y, \delta_X + \delta_Y\right)}$$

$$\frac{X : \left(\overline{\mathbb{E}}[X], \varepsilon_X, \delta_X\right), Y : \left(\overline{\mathbb{E}}[Y], \varepsilon_Y, \delta_Y\right)}{X \times Y : \left(\overline{\mathbb{E}}[X]\overline{\mathbb{E}}[Y], \varepsilon_X \varepsilon_Y + \overline{\mathbb{E}}[X]\varepsilon_Y + \overline{\mathbb{E}}[Y]\varepsilon_X, \delta_X + \delta_Y\right)}$$

$$\frac{X : \left(\overline{\mathbb{E}}, \varepsilon, \delta\right), \overline{\mathbb{E}} - \varepsilon > 0}{X^{-1} : \left(\overline{\mathbb{E}}^{-1}, \frac{\varepsilon}{\overline{\mathbb{E}}(\overline{\mathbb{E}}-\varepsilon)}, \delta\right)} \text{ (Inverse)} \qquad \frac{X : \left(\overline{\mathbb{E}}, \varepsilon, \delta\right)}{X^{-1} : \left(\overline{\mathbb{E}}^{-1}, \frac{\varepsilon}{\overline{\mathbb{E}}(\overline{\mathbb{E}}-\varepsilon)}, \delta\right), \{\overline{\mathbb{E}} - \varepsilon > 0\}} \text{ (Inverse Constr.)}$$

$$\frac{X : \left(\overline{\mathbb{E}}, \varepsilon, \delta\right), \overline{\mathbb{E}} - \varepsilon > c}{\psi \equiv X > c : (T, \delta)} \text{ (True)} \qquad \frac{X : \left(\overline{\mathbb{E}}, \varepsilon, \delta\right), \overline{\mathbb{E}} + \varepsilon < c}{\psi \equiv X < c : (F, \delta)} \text{ (False)}$$

$$\frac{X : \left(\overline{\mathbb{E}}, \varepsilon, \delta\right)}{\psi \equiv X > c : (T, \delta), \{\overline{\mathbb{E}} - \varepsilon > c\}} \text{ (True Constr.)}$$

$$\frac{X : \left(\overline{\mathbb{E}}, \varepsilon, \delta\right)}{\psi \equiv X < c : (T, \delta), \{\overline{\mathbb{E}} + \varepsilon < c\}} \text{ (False Constr.)}$$

$$\frac{\psi_1 : (\mathbb{B}_1, \delta_1), \psi_2 : (\mathbb{B}_2, \delta_2)}{\psi_1 \wedge \psi_2 : (\mathbb{B}_1 \wedge \mathbb{B}_2, \delta_1 + \delta_2)} \text{ (and)} \qquad \frac{\psi_1 : (\mathbb{B}_1, \delta_1), \psi_2 : (\mathbb{B}_2, \delta_2)}{\psi_1 \vee \psi_2 : (\mathbb{B}_1 \vee \mathbb{B}_2, \delta_1 + \delta_2)} \text{ (or)}$$

$$\frac{\psi_1 : (\mathbb{B}_1, \delta_1), \{C_{11,\ldots,1k}\}, \psi_2 : (\mathbb{B}_2, \delta_2), \{C_{21,\ldots,2m}\}}{\psi_1 \wedge \psi_2 : (\mathbb{B}_1 \wedge \mathbb{B}_2, \delta_1 + \delta_2), \{C_{11,\ldots,1k}, C_{21,\ldots,2m}\}} \text{ (and constr.)}$$

$$\frac{\psi_1 : (\mathbb{B}_1, \delta_1), \{C_{11,\ldots,1k}\}, \psi_2 : (\mathbb{B}_2, \delta_2)}{\psi_1 \vee \psi_2 : (\mathbb{B}_1 \vee \mathbb{B}_2, \delta_1 + \delta_2), \{C_{11,\ldots,1k}\} \vee \{C_{21,\ldots,2m}\}} \text{ (or constr.)}$$

Figure 5: Inference rules used to guarantees for expressions.The inference rules for each compound expression build on the union bound, triangle inequality, and structural induction approach described by Bastani et al. (2019).

**Inverse/Inverse constr.** Assume $X : (\overline{\mathbb{E}}, \varepsilon, \delta)$ and $\overline{\mathbb{E}} - \varepsilon > 0$. Instead, in the constrained case, we start with only the prior assumption i.e., $X : (\overline{\mathbb{E}}, \varepsilon, \delta)$ Then,

$$
\begin{aligned}
|\mathbb{E}[X]| &= |\mathbb{E}[X] - \overline{\mathbb{E}}[X] + \overline{\mathbb{E}}[X]| \\
&\leq |\mathbb{E}[X] - \overline{\mathbb{E}}[X]| + |\overline{\mathbb{E}}[X]| \\
&\leq \varepsilon_X + |\overline{\mathbb{E}}[X]|
\end{aligned}
$$

i.e., $|\mathbb{E}[X]| \leq \varepsilon_X + |\overline{\mathbb{E}}[X]|$. Also,

$$
\begin{aligned}
|\mathbb{E}[X]^{-1} - \overline{\mathbb{E}}[X]^{-1}| &= \left| \frac{\overline{\mathbb{E}}[X]^{-1} - \mathbb{E}[X]^{-1}}{\overline{\mathbb{E}}[X]\mathbb{E}[X]^{-1}} \right| \\
&\leq \frac{\varepsilon}{|\mathbb{E}[X]||\overline{\mathbb{E}}[X]|} \\
&\leq \frac{\varepsilon}{|\mathbb{E}[X]|(\mathbb{E}[X] - \varepsilon_X)}
\end{aligned}
$$

where the last step follows from the previous derivation and if $\mathbb{E}[X] - \varepsilon_X > 0$. The latter condition enforces that the sign of the inequality does not change. VF adds this as a precondition; we add it as a post-constraint.

**Boolean Operators** Starting from $\psi_1 : (b_1, \delta_1)$, $\psi_2 : (b_2, \delta_2)$, we can apply the union bound for $\psi_1 \wedge \psi_2$, $\psi_1 \vee \psi_2$ to derive the rules for and/or. Similarly, constraints follow the semantics specified by the rules as they also follow from the union bound.

## A.2 INFERRED OPTIMIZATION PROBLEM

For a given overall specification $\psi$, suppose $(\varepsilon_i, \delta_i)$, $i \in \{1, \ldots, n\}$ represents the concentration bounds associated with each constituent elementary subexpression. Using the aforementioned inference rules, we can derive the overall $\delta_T = \sum_i \delta_i$, along with a set of (say) $K$ constraints

$$
g_k(\varepsilon_1, \ldots, \varepsilon_n, \overline{\mathbb{E}}[X_1], \ldots, \overline{\mathbb{E}}[X_n]) \leq \varepsilon_k
$$

where

$$
\varepsilon_k = \left| c_k - \overline{\mathbb{E}}[f(\overline{\mathbb{E}}[X_1], \ldots, \overline{\mathbb{E}}[X_n])] \right|
$$

denotes the maximum allowed margin for the $k^{\text{th}}$ inequality subexpression (i.e. having form `<ETerm> <comp-op> c`). The objective is to minimize the overall failure probability $\delta_T$. The overall optimization problem can then be formulated as shown in 2, having $n$ optimization variables $\delta_i$ and $2n + K$ constraints (bounds on $\delta_i$ provide the $2n$ constraints). A developer using AVOIR inputs a required acceptable upper bound of failure probability $\Delta$. If the solution to the optimization problem $\delta_T^* = \sum_i \delta_i \leq \Delta$, then the optimization can conclude with the required confidence in the proved guarantee. At this point, the developer may choose to terminate AVOIR. However, using Corollary 4.1, they may continue to run and refine the estimates.

## B CONCENTRATION BOUNDS

The adaptive Hoeffding inequality (Zhao et al., 2016; Bastani et al., 2019).

**Theorem 3.** *Given a Bernoulli random variable X with distribution $P_X$. Let $\{X_i \sim P_X\}, i \in \mathbb{N}$ be i.i.d samples of X. Let*

$$
\overline{\mathbb{E}}_t[X] = \frac{1}{t} \sum_{i=1}^{t} X_i.
$$

*Let $\mathcal{T}$ be a random variable on $\mathbb{N} \cup \{\infty\}$ such that $\Pr[\mathcal{T} < \infty] = 1$, and let*

$$
\varepsilon(\delta, t) = \sqrt{\frac{\frac{3}{5} \log\left(\log_{1.1} t + 1\right) + \frac{5}{9} \log\left(24/\delta\right)}{t}}
$$

*Then, for any $\delta \in \mathbb{R}_+$, we have*

$$\Pr[|\overline{\mathbb{E}}_{\mathcal{T}}[X] - \mathbb{E}[X]| \leq \varepsilon(\delta, \mathcal{T})|] \geq 1 - \delta$$

.

Theorem 3 provides a mechanism for choosing the stopping time using arbitrary methods for a fixed $\delta$. Note that in general, any adaptive concentration inequality suffices; we use the Hoeffding inequality that does not depend on the empirical variance but is frequently used in scenarios dealing with bounded r.vs. However, we use confidence intervals to visualize the evolution of sub-expressions (and overall specification) over the sequence of observations. For doing so, we require an additional result

**Theorem 4.** *(Zhao et al., 2016, Proposition 1, Lemma 1) Let $S_n = \sum_{i=1}^{n} X_i$ be a random walk from i.i.d. random variables $X_1, \ldots, X_t \sim D$. For any $\delta > 0$,*

$$\Pr[S_{\mathcal{T}} \geq f(\mathcal{T})] \leq \delta$$

*for any stopping time $\mathcal{T}$ if and only if*

$$\Pr[\exists n, S_t \geq f(t)] \leq \delta$$

**Corollary 4.1.** *For any $\delta > 0$,*

$$\Pr[|\overline{\mathbb{E}}_{\mathcal{T}}[X] - \mathbb{E}[X]| \leq \varepsilon(\delta, \mathcal{T})|] \geq 1 - \delta$$

*for any stopping time $\mathcal{T}$ if and only if*

$$\Pr\left[\forall t, |\overline{\mathbb{E}}_t[X] - \mathbb{E}[X]| \leq \varepsilon(\delta, t)|\right] \geq 1 - \delta$$

*Proof.* Follows directly from applying Theorem 4 to Theorem 3. □

Intuitively, Theorem 3 holds since we can choose an adversarial stopping rule for $\mathcal{T}$ that terminates as soon as the boundary for $\varepsilon(\delta, t)$ is crossed (Zhao et al., 2016). Thus, when we establish a bound with a stopping rule, as long as the underlying distribution remains unchanged, the bound will hold prior to and after the stopping rule is enforced. Theorem 4.1 implies that once we choose an optimal bound for each subexpression, we can extend the bounds derived using Theorem 3 to following observations with the guarantees for the subexpressions still holding true.

## C   CONFIDENCE SEQUENCES

In this section, we show that the estimates generated from AVOIR form a confidence set (Theorem 1). First, we assume the existence of a concentration sequence for the mean of each elementary subexpression (eg., Theorem 3 can provide one). That is, we need a function $\varepsilon(t, \delta)$ such that

$$\Pr[\forall t \geq 1, |\overline{\mathbb{E}}_t[X] - \mathbb{E}[X]| \leq \varepsilon(t, \delta_X)] \geq 1 - \delta_X. \tag{5}$$

For convenience of exposition, we denote such adaptive inequality functions as AIN. For any AIN to be usable with AVOIR, we require $\varepsilon(t, \delta)$ to be monotonically non-increasing in $\delta$ and $n$. We expect this to be the case for most AIN, since increasing the number of observations of increasing the failure threshold should allow for additional concentration around the mean. For example, the adaptive Hoeffding inequality (Theorem 3) follows this assumption. Second, we assume that, except in degenerate cases, AVOIR terminates (see corollary 4.2 for termination criteria). We now prove Theorem 1.

*Proof.* First, we will prove that the estimates for *elementary* subexpressions are a confidence sequence. Following this, using the inference rules from Appendix A, we will show that the estimates for every compound expression are also a confidence sequence.

**Elementary subexpressions**  Consider a specification $\psi$ consisting of *elementary* subexpressions $X_1, \ldots, X_n$. At stopping time $\mathcal{T}$, let

$$\phi_{X_i}^{\mathcal{T}} := X_i : (\overline{\mathbb{E}}_{\mathcal{T}}[X_i], \varepsilon(\mathcal{T}, \delta_{X_i}), \delta_{X_i}) \tag{6}$$

be the stopping time estimates. Then, from the termination criterion, a solution to the optimization problem $\texttt{OPT}$ exists, i.e,

$$\Delta \geq \sum_i \delta_{X_i} \tag{7}$$

The sequence of bounds claimed by AVOIR are

$$\varepsilon_{X_i}(t) = \begin{cases} \varepsilon(\Delta, t), & t < \mathcal{T}, \\ \varepsilon(\delta_{X_i}, t), & t \geq \mathcal{T} \end{cases} \tag{8}$$

From equation 7 and the optimization constraint $\delta_i \in [0, 1]$ we have $\Delta \geq \delta_{X_i}$. From the non-decreasing behavior of AIN

$$\varepsilon(\Delta, t) \leq \varepsilon(\delta_i, t) \tag{9}$$

Now

$$\Pr[\forall t \geq 1, |\overline{\mathbb{E}}_t[X_i] - \mathbb{E}[X_i]| \leq \varepsilon_{X_i}(t)]$$

$$= 1 - \Pr[\exists t \geq 1, |\overline{\mathbb{E}}_t[X_i] - \mathbb{E}[X_i]| > \varepsilon_{X_i}(t)]$$

$$= 1 - \Pr\left[\bigcup_{t \geq 1} \left\{|\overline{\mathbb{E}}_t[X_i] - \mathbb{E}[X_i]| > \varepsilon_{X_i}(t)\right\}\right]$$

$$= 1 - \Pr\left[\bigcup_{t=1}^{\mathcal{T}-1} \left\{|\overline{\mathbb{E}}_t[X_i] - \mathbb{E}[X_i]| > \varepsilon_{X_i}(t)\right\} \cup \bigcup_{t \geq \mathcal{T}} \left\{|\overline{\mathbb{E}}_t[X_i] - \mathbb{E}[X_i]| > \varepsilon_{X_i}(t)\right\}\right]$$

(associativity of $\cup$)

$$= 1 - \Pr\left[\bigcup_{t=1}^{\mathcal{T}-1} \left\{|\overline{\mathbb{E}}_t[X_i] - \mathbb{E}[X_i]| > \varepsilon(\Delta, t)\right\} \cup \bigcup_{t \geq \mathcal{T}} \left\{|\overline{\mathbb{E}}_t[X_i] - \mathbb{E}[X_i]| > \varepsilon(\delta_{X_i}, t)\right\}\right]$$

(From 8)

$$= 1 - \Pr\left[\bigcup_{t=1}^{\mathcal{T}-1} \left\{|\overline{\mathbb{E}}_t[X_i] - \mathbb{E}[X_i]| > \varepsilon(\delta_{X_i}, t) \cup |\overline{\mathbb{E}}_t[X_i] - \mathbb{E}[X_i]| \in (\varepsilon(\Delta, t), \varepsilon(\delta_{X_i}, t)]\right\} \cup \right.$$

$$\left. \bigcup_{t \geq \mathcal{T}} \left\{|\overline{\mathbb{E}}_t[X_i] - \mathbb{E}[X_i]| > \varepsilon(\delta_{X_i}, t)\right\}\right] \text{ (Using 9)}$$

$$= 1 - \Pr\left[\bigcup_{t=1}^{\mathcal{T}-1} \left\{|\overline{\mathbb{E}}_t[X_i] - \mathbb{E}[X_i]| \in (\varepsilon(\Delta, t), \varepsilon(\delta_{X_i}, t)]\right\} \cup \bigcup_{t \geq 1} \left\{|\overline{\mathbb{E}}_t[X_i] - \mathbb{E}[X_i]| > \varepsilon(\delta_{X_i}, t)\right\}\right]$$

(Rearranging)

$$\geq 1 - \Pr\left[\bigcup_{t \geq 1} \left\{|\overline{\mathbb{E}}_t[X_i] - \mathbb{E}[X_i]| > \varepsilon(\delta_{X_i}, t)\right\}\right]$$

$$= 1 - \Pr\left[\exists t \geq 1, |\overline{\mathbb{E}}_t[X_i] - \mathbb{E}[X_i]| > \varepsilon(\delta_{X_i}, t)\right]$$

$$\geq 1 - \delta_{X_i}$$

where the last step follows from the definition of the adaptive concentration bound used. Thus, $\varepsilon_{X_i}(t)$ defines a $1 - \delta_{X_i}$ confidence sequence for $\mathbb{E}[X_i]$.

**Compound subexpressions**   Consider a non-specification compound `<ETerm>` $C_j$ consisting of *elementary* subexpressions with indices $C_j = \{\{j_1, j_2, \ldots, j_M\}\}$ as the decision r.v.s, i.e, $X_{j_1} \ldots, X_{j_M}$. Note that $C_j$ is a multiset as the same expression could occur multiple times within $C_j$. At stopping time $\mathcal{T}$,

$$\phi_{C_j}^{\mathcal{T}} : (\overline{\mathbb{E}}_{\mathcal{T}}[C_j], \delta_{C_j}, \varepsilon_{C_j}) \tag{10}$$

where $\overline{\mathbb{E}}_{\mathcal{T}}[C_j], \delta_{C_j}, \varepsilon_{C_j}$ are the corresponding values computed through the inference rules. In general, we denote by

$$\overline{\mathbb{E}}_t[C_j], \delta_{C_j}(t), \varepsilon_{C_j}(t) = \mathrm{INFER}(\phi_{X_{j_1}}^{t}, \ldots, \phi_{X_{j_M}}^{t}) \tag{11}$$

the values inferred at time step $t$, where INFER denotes the inference rules. Now,

$$\Pr[\exists t \geq 1, |\mathbb{E}[C_j] - \overline{\mathbb{E}}[C_j]| > \varepsilon_{C_j}(t)]$$

$$\leq \Pr\left[\bigcup_{i=1}^{M} \exists t \geq 1, \neg\phi_{X_{j_i}}^{t}\right] \qquad\qquad \text{(From 11)}$$

$$\leq \sum_{i \in C_j} \Pr\left[\exists t \geq 1, \neg\phi_{X_{j_i}}^{t}\right] \qquad\qquad \text{(union bound)}$$

$$= \sum_{i \in C_j} \Pr\left[\exists t \geq 1, |\overline{\mathbb{E}}_t[X_{j_i}] - \mathbb{E}_t[X_{j_i}]| > \varepsilon_{X_{j_i}}(t)\right] \qquad\qquad \text{(definition of } \phi_{X_{j_i}}^{t}\text{)}$$

$$\leq \sum_{i \in C_j} \delta_{X_{j_i}} \qquad\qquad \text{(elementary subexpressions)}$$

$$\leq \delta_{C_j} \qquad\qquad \text{(applying 11 for } t = \mathcal{T}\text{)}$$

Therefore $\varepsilon_{C_j}(t)$ defines a $1 - \delta_{C_j}$ confidence sequence for $\mathbb{E}[C_j]$

A similar proof can be constructed for any `<spec>`. Consider any specification $\psi_k$. Let

$$\psi_k^t : (\hat{b}_{\psi_k}(t), \delta_{\psi_k}(t)) \tag{12}$$

where $\hat{b}_{\psi_k}(t) \subseteq \{T, F\}$ is the inferred value and $\delta_{\psi_k}(t)$ corresponds to the confidence for the assertion at time $t$. Let the *elementary* subexpressions involved be $X_{k_1}, \ldots, X_{k_D}$ corresponding to the index multiset $B_k = \{\{k_1, \ldots, k_D\}\}$. Denote $b_{\psi_k}$ as the true value of $\psi_k$, and $\delta_{\psi_k}$ as the inferred threshold at stopping time $\mathcal{T}$. From INFER, we have

$$\hat{b}_k(t), \delta_{\psi_k}(t) = \mathrm{INFER}(\phi_{X_{k_1}}^{t}, \ldots, \phi_{X_{k_D}}^{t}) \tag{13}$$

We have

$$\Pr[\exists t \geq 1, b_k \notin \hat{b}_k(T)]$$

$$\leq \Pr\left[\bigcup_{i=1}^{D} \exists t \geq 1, \neg\phi_{X_{k_i}}^{t}\right] \qquad\qquad \text{(From 13)}$$

$$\leq \sum_{i \in B_k} \Pr\left[\exists t \geq 1, \neg\phi_{X_{k_i}}^{t}\right] \qquad\qquad \text{(union bound)}$$

$$= \sum_{i \in B_j} \Pr\left[\exists t \geq 1, |\overline{\mathbb{E}}_t[X_{k_i}] - \mathbb{E}_t[X_{k_i}]| > \varepsilon_{X_{k_i}}(t)\right] \qquad\qquad \text{(definition of } \phi_{X_{k_i}}^{t}\text{)}$$

$$\leq \sum_{i \in B_j} \delta_{X_{k_i}} \qquad\qquad \text{(elementary subexpressions)}$$

$$\leq \delta_{\psi_k} \qquad\qquad \text{(applying 11 for } t = \mathcal{T}\text{)}$$

Thus, $b_{\psi_k}(t)$ is a $1 - \delta_{\psi_k}$ confidence sequence for $b_{\psi_k}$ $\qquad\qquad\qquad \square$

## D   OPTIMALITY

*Proof.* Under $A_\delta$, at the stopping time $\mathcal{T}^+$, $\delta_i^+ = \Delta/n$, with

$$\sum_{i=1}^{n} \delta_i^+ = \Delta.$$

As $\delta_i^+$ are propagated using INFER (without constraint rules), we know that they must satisfy the constraints of the optimization problem 2. At time $\mathcal{T}^+$ AVOIR would find solution $\delta_i^*$ such that minimizes $\sum_{i=1}^{n} \delta_i$.

$$\sum_{i=1}^{n} \delta_i^* \leq \sum_{i=1}^{n} \delta_i^+ = \Delta$$

Thus, AVOIR would terminate at step $\mathcal{T}^+$, but may find a feasible solution at an earlier step, i.e. $\mathcal{T} \leq \mathcal{T}^+$.

$\square$

**Corollary 4.2.** *Under mild conditions, AVOIR terminates in finite steps with an assertion over the required specification.*

*Proof.* We know that the stopping time $\mathcal{T} \leq \mathcal{T}^+$, the stopping time for AVOIR. Thus, AVOIR would terminate whenever Verifiar can. For completeness, we provide the conditions under which Verifair terminates. Note that $c \in \mathbb{R}$ corresponds to a constant threshold involved in specification, also presented in the grammar and bound proagation rules.

> Clarified c

- For every subexpression $C_k$ occurring in the specification such that it is involved in the inverse or inverse constr. rules (i.e., $\overline{\mathbb{E}}[C_k]^{-1}$), $\overline{\mathbb{E}}[C_k] \neq 0, C_k \neq 0$

- For every subexpression $C_k$ such that it occurs a True/False type inequality (such as $C_k > c$), $\overline{\mathbb{E}}[C_k] \neq c, C_k \neq c$

$\square$

# E  IMPLEMENTATION

We built a python library to create specifications that can be implemented as a decorator over decision functions. The front end interactive application was implemented using streamlit[6] and the visualizations were built in Vega Satyanarayan et al. (2015). Each term in the DSL is implemented through a corresponding python class. New input/output observations are monitored to update all the terms in a specification. Inference for evaluating the value and bounds is carried out via operator overloading in these classes. In line with previous work (Albarghouthi et al., 2017; Bastani et al., 2019; Albarghouthi & Vinitsky, 2019) on distributional verification, we use rejection sampling for conditional probability estimation.

## E.1  VISUAL ANALYSIS

Using our specification framework as a backend, we built an interactive application for analysis and refinement of specifications provided in our grammar. Given a user provided machine learning model, dataset, and specification the application simulates a stream of observations to the provided model. Following the simulation, a visualization is provided that represents the specification as a syntax tree where each node of the tree corresponds to an element of our grammar. Figure 6 shows the visualization.

Note that for each observation made by our machine learning model, the specification is evaluated to check for violations. Each grammar element that makes up the specification is evaluated as well, and thus each grammar element is associated with the value it evaluates to for a given observation. For specifications `<spec>`, there is a boolean value associated with each observation, whereas an expectation term, `<ETerm>`, is associated with a real value. By selecting one of the nodes in the syntax tree, a user can see a plot of the evaluation values associated with the selected grammar element. We call these plots evaluation plots and two can be observed at a time each with shared scales along the horizontal axis which denotes observations over time. This allows for comparison

---

[6]https://streamlit.io/

---

**Algorithm 1** AVOIR Algorithm

---

**Input:** $\Delta, \psi$                                                 $\triangleright \Delta$, Specification
**Output:** $T_s$ time step when the value of $\psi$ can be guaranteed with probability $\geq 1 - \Delta$

1: **for** $X_i \in \psi$ **do**
2:      $\delta_{X_i} = \Delta$                                            $\triangleright$ Set initial value $\forall i$
3:      $S_{X_i} = 0$                                        $\triangleright$ Sum of observations
4:      $n_{X_i} = 0$                                     $\triangleright$ Number of observations
5: **end for**
6: $T = 0$                                               $\triangleright$ Time step
7: Initialize $OPT_\psi$                         $\triangleright$ Initialize Optimization Problem (Fig. 5)
8: **procedure** OBSERVE($X$)
9:      **for** $X_i \in X$ **do**
10:          $S_{X_i} = S_{X_i} + X_i$
11:          $n_{X_i} = n_{X_i} + 1$
12:          $\overline{\mathbb{E}}[X_i] = S_{X_i}/n_{X_i}$
13:          Initialize $\delta_{X_i}$ as a symbolic variable
14:          Assign $\varepsilon(\delta_{X_i}, n_{X_i})$ symbolic variable
15:      **end for**
16:      Propagate $\delta_{X_i}$ using the inference rules
17:      Initialize constraints $g_K$ in $OPT_\psi$ using the computed values
18:      $\delta_T^* = \texttt{Solve}(OPT_\psi)$
19:      **if** $\delta_T^* \leq \Delta$ **then**
20:          $\delta_{X_i} = \delta_T^*[X_i]$
21:          **return** $T_s = T$
22:      **end if**
23:      $T = T + 1$
24: **end procedure**

---

of multiple grammar elements. The ability to analyze and compare these evaluation values provides context surrounding specification violations, and assists the user in deciding how to refine a specification. The case studies in section 3 demonstrate the usefulness of the context provided by these visualizations.

## F   AVOIR IN DATABASE SETTING

In the database literature researchers Nargesian et al. (2021), have explored an approach to tailoring data integration strategies to ensure that the data set used for analysis has an appropriate representation of relevant (demographic) groups and it meets desired distribution requirements. The authors describe how to acquire such data in an approximate cost-optimal manner for several realistic settings. This work is orthogonal to our work and yet AVOIR can potentially integrate with the authors approach to examine if fairness criteria are being met during the integration process. In other studies on fairness researchers Yang et al. (2018); Asudeh et al. (2019); Sun et al. (2019), have considered the problem of personalized fair ranking functions and discuss approaches to determine if a proposed ranking function satisfies a set of desired fairness criteria and, if it does not, to suggest modifications that do. AVOIR attempts to solve a more general purpose problem (not limited to any particular fairness criteria) and is agnostic to the specific model (treats it as a blackbox). While we have not examined the performance of AVOIR for fair ranking problems, it is something we plan to examine in the future.

To demonstrate how AVOIR can be integrated within a database system we use pandas[7] dataframes to simulate the application of AVOIR in the database setting. Specifically, we wrap pandas dataframes with a python 'Database' class, and provide a query mechanism to create materialized views. Queries are provided in the form of python functions that take a dataframe as input and output a corresponding dataframe. The corresponding view thus generated can be updated with insertion/update/deletion of data. The specification is added as a decorator inside the refresh function,

---

[7]https://pandas.pydata.org/

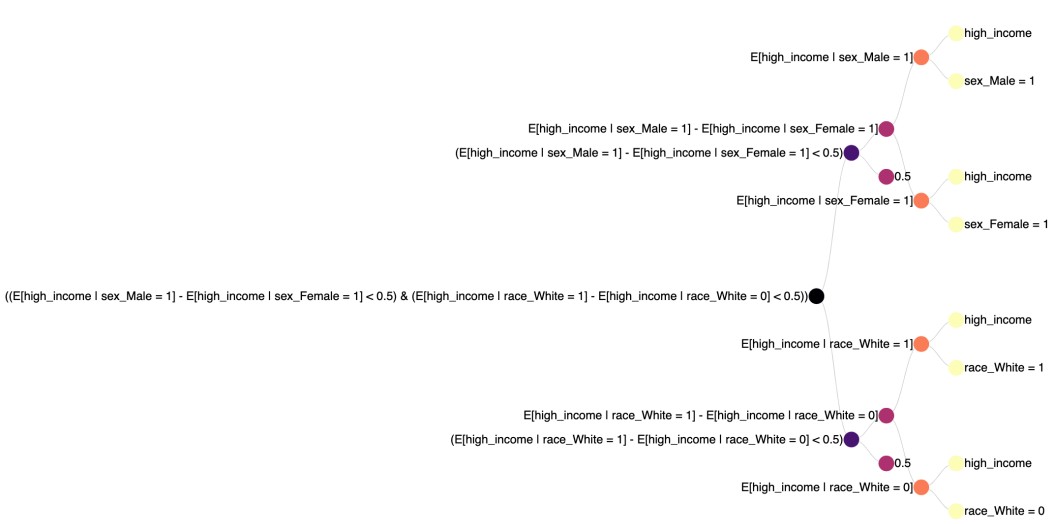

Figure 6: Tree of initial specification before refinement in the adult income dataset.

allowing AVOIR to track specifications in a database setting. Note that this tie-in with pandas is only for ease of implementation; the inference engine and optimization can be extended to any database engine.

## G  ADDITIONAL CASE STUDIES

### G.1  MATERIALIZED VIEWS

A materialized view is constructed by querying the dataset to select for employees of the federal government. We simulate the materialized view using a pandas[8] dataframe wrapped in a python class to monitor updates and run AVOIR for any monitored specification.

### G.2  INTERACTION THROUGH VEGA

Figure 6 shows a subtree of the specification visualized through Vega. A developer analyzing this spec can click on the top pink node to see the evolution of the sex fairness part of the specification and superimpose the threshold. The threshold is set to be evaluated with every 5 new data points added to the materialized view. Clicking on the corresponding element in the right subtree, the developer can see Figure 4b.

### G.3  COMPAS RISK ASSESSMENT VIA MATERIALIZED VIEWS

The Correctional Offender Management Profiling for Alternative Sanctions (COMPAS) recidivism risk score data is a popular dataset for assessing machine bias of commercial tools used to assess a criminal defendant's likelihood to re-offend. The data is based on recidivism (re-offending) scores derived from a software released by Northpointe and widely used across the United States for making sentencing decisions. In 2016, Angwin et al. (2016) released an article and associated analysis code critiquing machine bias associated with race present in the COMPAS risk scores for a set of arrested individuals in Broward County, Florida over a period of two years. The analysis concluded that there were significant differences in the risk assessments of African-American and Caucasian individuals. Northpointe pushed back in a report (Dieterich et al., 2016) strongly rejecting the claims made by the ProPublica article; instead, they claimed that Angwin et al. (2016) made several statistical and

---

[8]https://pandas.pydata.org/

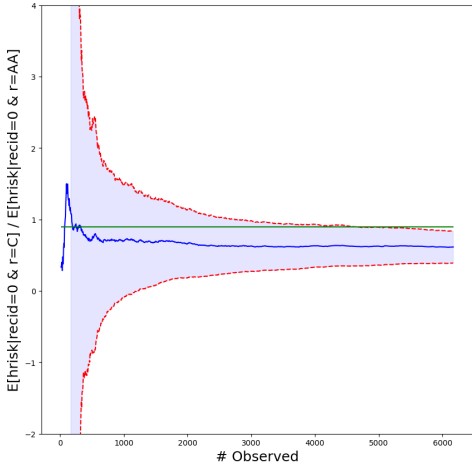
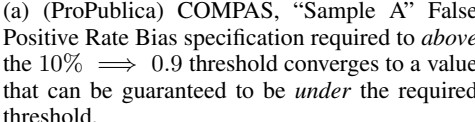
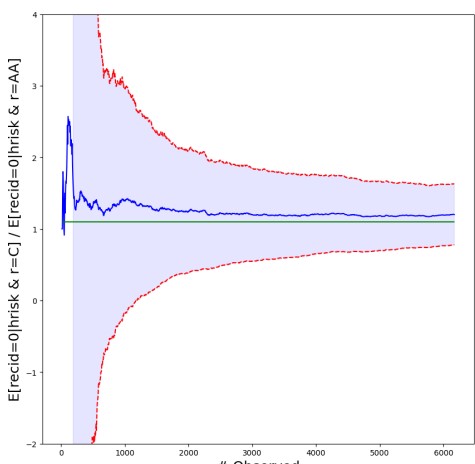

(a) (ProPublica) COMPAS, "Sample A" False Positive Rate Bias specification required to *above* the 10% $\implies$ 0.9 threshold converges to a value that can be guaranteed to be *under* the required threshold.

(b) (Northpointe) "Sample B" analysis done by Northpointe using False Discovery Rate that opposed the ProPublica reports.

Figure 7: COMPAS dataset case study.

technical errors in the report. In this case study, we use AVOIR to study the claims made by the two aforementioned reports. First, we start with the data released by ProPublica and load it into a pandas-simulated DB. We then create a materialized view that corresponds to the preprocessing steps used in the publicly available ProPublica analysis notebook[9]. We look at "Sample A" (Dieterich et al., 2016), where the analysis of the "not low" risk assessments using a logistic regression model reveals a high coefficient associated with the factor associated with race being African-American. In terms of a fairness metric, this corresponds to false positive rate (FPR) balance (predictive equality) (Verma & Rubin, 2018) metrics. The associated specification in AVOIR grammar would be

E[ hrisk | race=African−American & recid=0] /
E[ hrisk | race=Caucasian & recid=0] < 1.1

Where `hrisk` is an indicator for high risk assessments made by the *black-box* COMPAS tool as defined by Angwin et al. (2016), `recid` is an indicator for re-offending within 2 years of first arrest, and a 10%-rule is used as the threshold. We choose a failure threshold probability of $\Delta = 0.1$, with the optimization run after every batch of 5 samples. AVOIR finds that when the decisions are made in a sequential, online fashion, the assertion for violation of the specification cannot be made with the required failure guarantee.

By analyzing the components using the visualization tool, one can glean that AVOIR is unable to optimize since the lower FPR in the denominator (FPR for Caucasian individuals) increasing the overall variance and limiting the ability to optimize for guarantees. We follow this analysis by using the reciprocal specification, i.e.,

E[ hrisk | race=Caucasian & recid=0] /
E[ hrisk | race=African−American & recid=0] > 0.9

we find that indeed, the specification is violated with a confidence of over $1 - \Delta = 0.9$ probability, and AVOIR can detect this violation within about half the number of available assessments (3350 steps) when run in an online setting. Figure 7a demonstrates the progression of the tracked expectation term. Thus, if deployed with the corrected specification, AVOIR would be able to alert Northpointe of a violation/potentially-biased decision making tool.

The Northpointe report (Dieterich et al., 2016) makes several claims about the shortcomings of this analysis, but one of the primary claims is that Angwin et al. (2016) used an analysis based on "Model

---

[9]https://github.com/propublica/compas-analysis

| Metric Name | Definition | DSL |
|---|---|---|
| Statistical Parity (Dwork et al., 2012) | $\Pr[d=1\|G=m] = \Pr[d=1\|G=f]$ | $\mathbb{E}[d\|G=m]/\mathbb{E}[d\|G=f] < c$ |
| Predictive Parity (Chouldechova, 2017) | $\Pr[Y=1\|d=1,G=m] = \Pr[Y=1\|d=1,G=f]$ | $E[Y=1\|d=1,G=f] - \mathbb{E}[Y=1\|d=1,G=m] > c$ |
| Equal Opportunity (Hardt et al., 2016) | $\Pr[d=0\|Y=1,G=m] = \Pr[d=0\|Y=1,G=f]$ | $\mathbb{E}[d=0\|Y=1,G=m] - \mathbb{E}[d=0\|Y=1,G=f] < c$ |
| Equalized Odds (Hardt et al., 2016) | $\Pr[d=1\|Y=i,G=m] = \Pr[d=1\|Y=i,G=f]$, $i=0,1$ | $(\mathbb{E}[d=1\|Y=1,G=f] - \mathbb{E}[d=1\|Y=1,G=m] > c_1) \& (\mathbb{E}[d=1\|Y=0,G=f] - \mathbb{E}[d=1\|Y=0,G=m] > c_2)$ |

Table 1: Examples of supported metrics.

Errors" rather than "Target Population Errors". In Fairness metric terms, this refers to the difference between a False Positive Rate (FPR) balance vs False Discovery Rate (FDR) balance, i.e. balancing for predictive parity over predictive equality. In probabilistic terms, the difference amounts to comparing $P(\hat{Y}=1|Y=0, g=1,2)$ (FPR) vs $P(Y=0|\hat{Y}=1, g=1,2)$ (FDR), where $\hat{Y}$ refers to the decision made by the algorithm, $Y$ refers to the true value, and $g=1,2$ reflects group membership (Verma & Rubin, 2018). This analysis is run on the dataset subset dubbed "Sample B". To test their hypothesis, we run reproduce the corresponding preprocessing steps and run both versions (numerator and denominator being Caucasian) versions of the corresponding specification under the same setup as earlier. We find that despite the point estimate being within the required threshold, neither version can be guaranteed with the required confidence with the given data. Due to paucity of space, we describe only one of the two variants with the corresponding figure (Figure 7b).

```
E[recid=0 | race=Caucasian & hrisk] /
E[recid=0 | race=African−American & hrisk] > 0.9
```

We note that the Northpointe report (Dieterich et al., 2016) does not provide confidence intervals for their claim either. Further, even though the report does not release associated code, the point estimates of the False Discovery Rates (FDRs) match those present in the report which increases our confidence in our AVOIR-based analysis.

The back and forth exchange has been the subject of much discussion in both academic and journalistic publications (Feller et al., 2016; Washington, 2018). Seminal work by Kleinberg et al. (2017) proved the impossibility of simultaneously guaranteeing certain combinations of fairness metrics. While AVOIR cannot solve this problem, its usage can help provide explicit guarantees on defined metrics. The specification grammar also provides a simple mechanism for independent replication of claims. We conclude this case study by noting that AVOIR lends itself to successful analysis that is not possible with the Verifair implementation available online.

## H  SUPPORTED METRICS

We provide a non-exhaustive list of statistical group-based fairness criteria and show an exact/approximate equivalent in the AVOIR DSL in Table 1. We use the following notation, adapted from Verma & Rubin (2018):

- $G$: Protected or sensitive attribute. For demonstration purposes, we will use the values $m$ and $f$ to denote majority and minorty classes.

- $X$: Features describing each individual

- $Y$: True label for $X$

- $S$: Probability $\Pr[Y|X,G]$ predicted for a certain class $c$

- $d$: Predicted decision for $X$, usually derived from $X$

- $c$: A threshold to test the specification. For ratios based approximations, this would be a number $1 \pm \varepsilon$ for some small $\varepsilon > 0$. For difference based approxiamtions, this number would be some small $\varepsilon > 0$. When multiple terms are present, we use $c_i$ to denote the $i^{\text{th}}$ threshold.

New section

We assume that the decision function $f$ tracked by AVOIR as a signature that takes $X, G, Y$ as input and produces $S$ or $d$ as output. Note that in their python implementation, $=$ would be replaced by $==$ and $|$ by the given keyword.

