# OpenReview forum: "Auditing Fairness Online through Interactive Refinement"
_ICLR.cc/2023/Conference — Submitted to ICLR 2023_

### Official Review · Reviewer_Uuem · 2022-10-25

**Confidence:** 4
**Correctness:** 2
**Technical Novelty And Significance:** 2
**Empirical Novelty And Significance:** 3
**Recommendation:** 3

**Clarity, Quality, Novelty And Reproducibility:**

Please check the weakness for comments about clarity and quality. Novelty is not explicit from the methodological description. It is only visible from the comparison with the related work.

**Strength And Weaknesses:**

Strength
- The paper addresses an interesting problem of auditing fairness automatically, online, and in a black-box manner. This extends the previous fairness auditors by one or more extents.
- The paper presents an extensive summary of the related works and points the gaps where they can be improved.
- The paper performs elaborate case studies and develop a tool to visualise the results.


Weakness and questions:
- The paper suffers from poor readability. The core contribution is not properly stated. More focus has been paid to contrasting the work with existing literature while failing to convey a coherent story.
- It is not clear why the number of iterations is less for the proposed framework than the existing work, VeriFair.
- In Sec 2.2, what is F?
- Section 2.3.1 is written very poorly. There is no concrete mathematical statement to follow. What do you mean by sub-expression? What do you mean by Bernoulli random variables X_{1, 2}? What is the implication of the subscripts {1, 2}? What is the definition of failure probability?
- Can you elaborate on this statement: "We can claim stronger guarantees for X 2 if t 2 > t 1 as the failure probability is lower at the same concentration"?
- In Sec 3.3.2, while no solution is feasible for the optimization problem with A_\delta, AVOIR finds a solution. What is the reasoning? What is A_\delta here?

**Summary Of The Paper:**

The paper investigates the problem of auditing fairness automatically, online, and in a black-box manner. It proposes AVOIR to address this problem for multiple fairness metrics. It claims to optimize the previously used adaptive Hoeffding inequality to decrease the sample complexity. AVOIR also comes along a visualisation and computation schema to compute and communicate the fairness violation. The paper also presents case studies on real-life datasets to justify its usefulness.

**Summary Of The Review:**

The paper addresses an interesting and timely problem of auditing fairness automatically, online, and in a black-box manner. Though the paper spends a lot of effort to compare with the related work, it does not concretely state its contributions (specifically sec 2.3.1 is poorly written). I think that the paper should be rewritten and polished to explicate the contributions with concrete mathematical statements. Otherwise, it is presently hard to evaluate the methodology now.

---

> ### Author Response · Authors · 2022-11-16
> **Response to Reviewer Uuem**
>
> We thank the reviewer for their insightful comments. We respond below:
>
> > The paper suffers from poor readability. The core contribution is not properly stated.
>
> Based on reviewer feedback we recognize that the table oversimplified the comparison with prior work and the original introduction was not clear and has been a consistent source of confusion. We have removed this and instead added a much more comprehensive related work section (1.2). We have also added a section on preliminaries to improve the readability. In addition, we have added sections to clarify the different terms (Section 1.1) and our contributions (in the [overall response](https://openreview.net/forum?id=Gp91Et4LeRf&noteId=hx4RuvvITrh) as well as Section 1.3 in the paper).
>
> > It is not clear why the number of iterations is less for the proposed framework than the existing work, VeriFair
>
> Theorem 2 in the paper shows how within AVOIR-OB, there are fewer constraints on the solution, and hence it can potentially terminate in fewer iterations than AVOIR-VF. This is empirically verified on the RateMyProfs and Adult Income datasets.
>
> Additionally, Verifair assumes that the data distribution is known and can be sampled from. To deal with unknown distributions, Verifair requires an additional fitting step of a density estimation model. The confidence set framework and inference rules in AVOIR enable AVOIR-VF to sidestep this assumption making AVOIR-VF more efficient than the original Verifair approach.
>
> > In Sec 2.2, what is F?
>
> T/F refer to True and False. We have also updated our writing to clarify this.
>
> > Section 2.3.1 is written very poorly. There is no concrete mathematical statement to follow. What do you mean by sub-expression?
>
> As stated earlier, we have added a preliminaries section (1.1) and also expanded the language specification section (2.1) to clarify the definitions of different terms.
>
> > Can you elaborate on this statement: "We can claim stronger guarantees for X 2 if t 2 > t 1 as the failure probability is lower at the same concentration"?
>
> We replaced “stronger” with “tighter” to be more precise, and updated the language around this statement to reflect this. Broadly, the claim is that if one sees more samples of one class than the other, the majority class will be more likely to be concentrated within a bound around the empirical estimate.
>
> > In Sec 3.3.2, while no solution is feasible for the optimization problem with A_\delta, AVOIR finds a solution. What is the reasoning? What is A_\delta here?
>
> $A_\delta $ is the assumption common to prior work $\delta_i = \delta_j$. We have rearranged this section to introduce $A_\delta$ earlier in this section.

---

### Official Review · Reviewer_Guzx · 2022-10-26

**Confidence:** 2
**Correctness:** 4
**Technical Novelty And Significance:** 2
**Empirical Novelty And Significance:** 2
**Recommendation:** 5

**Clarity, Quality, Novelty And Reproducibility:**

**Clarity** The paper is hard to understand for readers without the appropriate background knowledge, and there is no related work or background knowledge presented in the paper.


** Quality** The submission is technically sound.


** Novelty** The work seems to be an incremental combination of well-known techniques.


**Reproducibility** The paper contains enough details (e.g., codes) to reproduce the results.

**Strength And Weaknesses:**

Strengths:
1. The paper studies the fairness audition problem, which is of high importance.

2. AVOIR system is easy to integrate with the existing ML system with arbitrary fairness metrics.


Weaknesses

1. The paper’s presentation needs significant improvements to allow readers outside the fairness audition area to understand. While reading the paper, I had a hard time understanding the technical details of the proposed algorithm and the experimental results. This area contains a lot of background information from software testing, and a lot of the paper’s results are based on Albarghouthi & Vinitsky (2019). Besides, the paper does not provide a background information section or related work section, making it more difficult to understand the technical details.

2. The Novelty needs to be clarified. In table 1, compared to all other works, the only advantage of the system is visual refinement (e.g., tree visualization). In this regard, the contribution is not that significant. Besides, I found that the framework proposed in the paper is similar to Albarghouthi & Vinitsky (2019). According to Table 1, AVIOR shows advantages over Albarghouthi & Vinitsky (2019) regarding Adaptive Optimization and Visual Refinement. However, adaptive optimization has been integrated into Ginart et al. (2022) and Ghosh et al. (2021a). If visual refinement is only the improvement over the previous work, then the contributions are incremental.

**Summary Of The Paper:**

The paper presents AVOIR, a fairness audit framework that allows the generation of fairness probabilistic guarantees for any models and fairness metrics. AVOIR also uses a tree-based visualization to help users better the fairness specification.

**Summary Of The Review:**

The paper needs improvement in its presentation and clear justification of the novelty.

---

> ### Author Response · Authors · 2022-11-16
> **Response to Reviewer Guzx**
>
> We thank the reviewer for their insightful comments. Our response:
>
> > W1 The paper’s presentation needs significant improvements.
>
> > W2: The Novelty needs to be clarified. $\dots$ However, adaptive optimization has been integrated into Ginart et al. (2022) and Ghosh et al. (2021a). If visual refinement is only the improvement over the previous work, then the contributions are incremental.
>
> To improve our presentation, we have removed Table 1.1 which oversimplified the contributions and instead added a much more comprehensive related work section (1.2). We have also added a section on preliminaries to clarify the different terms (Section 1.1) and clarified our contributions in the overall response as well as Section 1.3 in the paper. We distinguish between adaptive optimization in general (eg. Ginart et al) monitor accuracy, Ghosh et al only apply to decision trees and linear models).
>
> **Novelty**
>
> We clarify this more broadly in our [overall response](https://openreview.net/forum?id=Gp91Et4LeRf&noteId=hx4RuvvITrh), and have also added a more fleshed-out contributions section (1.3)  to reflect the scope of our contributions more accurately. To reiterate, the key differentiators of AVOIR are
> * AVOIR relies on adaptive optimization via a unique confidence set framework
> * More efficient Bound Propagation
> * Inference Engine.

---

### Official Review · Reviewer_1wit · 2022-11-02

**Confidence:** 2
**Correctness:** 4
**Technical Novelty And Significance:** 2
**Empirical Novelty And Significance:** 2
**Recommendation:** 5

**Clarity, Quality, Novelty And Reproducibility:**

Clarity:
- Parts of the paper are very clear, e.g., the bound propagation and language specification sections. But what does "online" means in this setting? The authors say that this is an online monitoring system---are we assuming new data from the same distribution is coming in at each iteration? Or is this related to steps in the optimization problem?
- Is the BERT-based transformer model (sec 4.1) trained with knowledge of the fairness metrics?

Originality: Related work needs to be fleshed out a little more for readers to understand where on the frontier this work is. The extension of the Fairness Aware Programming is minimal, and the union bound technique has similarities to other methods that provide probabilistic guarantees but the differences/similarities are missing. This is a problem because one of the larger contributions of this work is the optimization of the concentration bounds.

Reproducible: Yes, see sec 7.

Theory: I found no issues in Appendix A and B, but did not look at or skim Appendices C-F.

Small suggestions:
- Para before sec 3.3: "see Appendix A" AIN used in Appendix A but has yet to be defined in the main body.
- Appendix A second para: "ise ideas" --> "use ideas"
- 3.4 second to last line: "interactiving with" --> "interacting with"
- Why is Verma & Rubin 2018 being referenced for fairness defns (Appendix G.3)? The original source should be referenced. "Fairness defns explained" is an outline of fairness defns that already exist in the literature, and provides intuition for their meaning using a specific application as example. This work does not introduce a new fairness metric.

**Strength And Weaknesses:**

Strengths
- Many sections of the work are well-written, including the introduction and some proofs in the appendices (A and B). The examples provided throughout the text help build intuition, e.g., Figure 2.
- The bound propagation and optimization techniques seem useful, especially for problem settings in which high probability fairness guarantees are necessary.
- The visualization techniques proposed offer additional guidance to users that interested in determining the specific fairness metric for the application at hand.

Weaknesses
- Missing related work on techniques with probabilistic fairness guarantees. The only related work in this space mentioned is AVOIR, but there are numerous other techniques in fair ML that propose probabilistic fairness guarantees. For example, the Seldonian framework ('Preventing undesirable behavior of intelligent machines'; Thomas 2019) is a family of algorithms that provides high confidence fairness guarantees; and 'Offline contextual bandits with high pr guarantees' (Metevier 2019) introduces a framework that uses interval arithmetic & the union bound to produce tight(er) bounds on statistical metrics of interest.

**Summary Of The Paper:**

This work introduces a framework/interface that allows users to (1) define fairness-based rate constraints/metrics, and (2) monitor these metrics / work to refine the initial constraint definitions (if necessary). The work also provides case studies showcasing the framework.

1. Define fairness-based rate constraints/metrics. The grammar used in the framework is based on the one used in Fairness Aware Programming with the addition of two extensions: direct specification of conditional statements, and the addition of some binary operators. Moreover, probabilistic guarantees based on union bound and interval algebra are provided that provide tighter confidence bounds on statistical metrics of interest.

2. Monitor these metrics / work to refine initial constraint defns. The framework includes a way for users to see a visual of the statistical metrics that may be in violation of fairness. A case study is shown that details how this information can help users choose an appropriate fairness metric.

**Summary Of The Review:**

I recommend rejection pending clarifications from the authors.

Work related to the broader subject of high probability fairness guarantees seems to be missing from the text. As such, it is hard to determine how similar some of the key contributions are to prior work, and therefore hard to determine the novelty of the work. These contributions also seem to be the most substantial (e.g., the additions to the grammar and the visualization proposed are contributions but more minor in scope).

---

> ### Author Response · Authors · 2022-11-16
> **Response to Reviewer 1wit**
>
> We thank the reviewer for their comprehensive review. We address their comments below:
>
> > Missing related work on techniques with probabilistic fairness
>
> We thank the reviewer for indicating the work on the Seldonian framework and have added it to a more comprehensive related work section.
>
> **Clarity:**
>
> >Parts of the paper are very clear, e.g., the bound propagation and language specification sections. But what does "online" means in this setting? The authors say that this is an online monitoring system---are we assuming new data from the same distribution is coming in at each iteration?
>
> The data is assumed to be generated one data point/batch at a time  from a fixed, but unknown distribution. We also adaptively optimize the thresholds over the elementary subexpressions. Both these properties motivate us to use ‘online’ in naming our framework.
>
> > Is the BERT-based transformer model (sec 4.1) trained with knowledge of the fairness metrics?
>
> No, we do not specifically tune the transformer for fairness. Keymanesh et al, carry out certain transformations to modify the reviews to remove gender references, and AVOIR is able to provide a guarantee for a specified gender fairness criterion.
>
> > Small suggestions
>
> We thank the reviewer for their suggestions, and have corrected the typos as well as added individual references for fairness specifications (Section 1.1 and Appendix H)
>
> **Novelty**
>
> We clarify this more broadly in our [overall response](https://openreview.net/forum?id=Gp91Et4LeRf&noteId=hx4RuvvITrh), and have also added a more fleshed-out contributions section (1.3)  to reflect the scope of our contributions more accurately. To reiterate, the key differentiators of AVOIR are
> * AVOIR relies on adaptive optimization via a unique confidence set framework
> * More efficient Bound Propagation
> * Inference Engine.

---

### Official Review · Reviewer_nDjC · 2022-11-03

**Confidence:** 4
**Correctness:** 2
**Technical Novelty And Significance:** 3
**Empirical Novelty And Significance:** 2
**Recommendation:** 3

**Clarity, Quality, Novelty And Reproducibility:**

Clarity:

See weaknesses above. Unfortunately, the paper suffers from several clarity issues in terms of its organization, precision w.r.t. defining math-related terms in the main paper (this is improved in the appendix), and claims in relation to improvements over prior work that do not seem to be entirely substantiated (but perhaps are, making it an issue of clarity).

Quality:

Due to the clarity issues above -- particularly those pertaining to imprecision -- a fair evaluation of the quality is, in my opinion, not entirely possible. Perhaps the work is a substantial improvement over prior work. Perhaps it is not (see comment in weaknesses re: relationship to Bastani et al., as an example)

Novelty:

Doesn't seem entirely novel, but this perhaps is because the improvements over prior work (versus parts that rely on prior work) is a bit opaque (see weaknesses).

Reproducibility:

Perhaps reproducible, but there is an inadequate exploration of the empirical examples to actually (in my opinion) demonstrate iterative refinement. I would expect more significant evaluation of a work that claims utility for online monitoring.



**Strength And Weaknesses:**

Strengths:

1. The problem the authors have set out to solve is an important one. Having good tools for identifying runtime fairness violations in deployed systems is an important goal.

2. The authors acknowledge that this, however, is not enough. There needs to be a way to interact with / effectively debug violations to improve the power of auditing tools.

3. The authors also make the contribution that this can be done efficiently (at least, more efficiently than prior work).

Weaknesses:

1. Unfortunately, due to lack of specificity / a preliminaries section that adequately specifies the setup, it is not entirely clear what exactly the extent of the contribution is.  For example, 2.1 needs to be more precise. It would be useful to specify a set of fairness criteria that are in scope, rather than speaking generally about fairness criteria. Work on Rawlsian social welfare does not seem to be in scope for this work, yet is arguably in scope in terms of the generality of how fairness criteria are written about in 2.1. This makes 2.1 (strictly speaking) incorrect. A dedicated preliminaries section would help fix this.

2. More precision in the math terms would also be helpful. For example (1) should define (above or below) what the types are for the symbols being used. What is r? Should we consider s to be applicable only to binary settings, or to multi-group fairness settings? Is the example fairness criterion something that the paper actually will use? If not, why is it there? If the definitions that are in scope are those from Verma and Rubin, can (briefly) a set of such definitions be described rigorously, with an example metric related to that set?

3. I think the authors need to explain more as to why this work provides a significant improvement over Bastani, et al. Notably, in Table 1, the improvement of AVOIR seems to be visual refinement. But is that actually enough of an improvement / a contribution to make this paper novel? It's not entirely clear, as presented. A dedicated related work section (as opposed to peppering some related work into the intro, and some in section 2) would be useful, and them some comparison in terms of evaluation (to verify the claim of improved efficiency) would be useful here. Moreover, if visual refinement is to be evaluated as a significant contribution, this should be featured more heavily in the paper. There is not a single example of this in the main paper, and the one in the Appendix is not adequately explored/ explained to justify its utility.

4. 2.3.2 seems to contradict the claim about operating on Bernoulli r.v.'s. "Arbitrary" suggests that any metric could be used, but this is not true. This feels like a contradiction from a lack of precision.

5. I do not understand figure 3b. What does it mean to have Avoir-Verifiair? Verifair is talked about like a prior work baseline, but this suggests some combination? (ultimately, this became a bit clearer at the beginning of Section 4, but I lost the thread here and had to work to find it again, and I'm not sure I completely did).

6. What exactly is being interactively tuned? This is said at a high level and not entirely made precise, in my opinion. I had hoped to see perhaps one long empirical example of online refinement. Perhaps one in which distribution shift is induced, so that we can see how the system catches it quickly in relation to prior work (this would support the theoretical claims being made). Without one, it is not clear what the benefit of this system is (and this matters a lot, since there is a claim about runtime improvement).

Nits, suggestions (I believe the following would improve the paper, but they do not impact my score):

1a. The claims in the abstract and intro concerning accountability do not seem correct to me. The link between audits and accountability has been explored in various works, none of which are cited here. Importantly, this link is not a given (as it seems to be suggested here). Audits are not necessarily constitutive of accountability. The concrete takeaway here, for this framing to work effectively, is to defer to those that have studied this problem in depth, rather than claiming this as a given without substantiating it. For an example on the complicated relationship between accountability and auditing, I would read sections of and cite Helen Nissenbaum's update to her classic paper that was at FAccT last year, "Accountability in an Algorithmic Society" https://dl.acm.org/doi/abs/10.1145/3531146.3533150

1b. Related to the above, the paper makes claims in the introduction about the challenges of accountability for ML, and has a paragraph that talks about a subset of problems with ML that make it special/difficult in unique ways. These claims, as written, are not adequately substantiated. Again, many others study this as a problem in itself. It is worth deferring to this work via citations, and then highlighting the two examples given in the intro as the ones that are relevant to this paper. Again, see paper example in 1a (and associated related work citations in that paper).

2. Many typos. Please proofread. One example, "stochaistic" (mispelling) in intro, second to last paragraph.




**Summary Of The Paper:**

This paper designs a DSL for specifying/verifying probabilistic guarantees for Bernouilli rv-related fairness metrics in online settings. The authors claim that their approach improves over prior work in terms of having more efficient optimization.

**Summary Of The Review:**

The paper contains some interesting ideas, but it is not written very clearly and has some issues with precision that make it difficult to fully evaluate both the scope and correctness of the contributions, as well as to fairly evaluate why this work is a substantial improvement over prior work. It is possible that the main issue is lack of clarity in the writing. If that is the case, then a deep re-write would resolve most of my concerns. However, I do not think that is the limit of the problems. As noted above in weaknesses, I think the paper needs deeper empirical investigation/demonstration to really benefit 1) the contribution 2) support of the theoretical claims of better efficiency 3) comparison to prior work. Unfortunately, as the paper stands, I do not think it is of high enough quality to be accepted to a top-tier conference. The ideas are interesting and the paper seems to have the potential to pass this bar, but needs reworking in order to do so.

---

> ### Author Response · Authors · 2022-11-16
> **Response to Reviewer nDjC**
>
>
> We thank the reviewer for their comprehensive review, and especially for pointing us to relevant literature on linking accountability and auditing. We have updated our review to better reflect this connection. We also respond to their comments below:
>
> > W1/W2: a preliminaries section and precision with mathematical definitions.
>
> We recognize the oversimplification of certain nuances regarding definitions in the original version of the submission. Many of these arise from an assumption of shared terminology from related work. We have added a preliminaries section, and expanded the language specification section to clarify these. In addition, we have added Appendix H to demonstrate examples of AVOIR with various fairness metrics.
>
> We have also added additional references.
>
> > W3: significant improvement over Bastani, et al.
>
> Based on reviewer feedback we recognize that the table oversimplified the comparison with prior work and the original introduction was not clear. We have now removed the table and added details on how AVOIR broadly differs from related work (1.2). Additionally in our contributions section (1.3) we more specifically clarify the differences between AVOIR  and  closely related work from the fairness testing domain. We also address this issue more broadly in the [overall response](https://openreview.net/forum?id=Gp91Et4LeRf&noteId=hx4RuvvITrh) to all reviewers.
>
> > W4: "Arbitrary" fairness metrics
>
> We have corrected this in the updated section on key contributions (1.3), and added examples in Appendix H.
>
> >W5: What does it mean to have Avoir-Verifiair?
>
> We have clarified this in the overall response and also in the key contributions section. The implementation of the bound propagation rules in our inference engine allows the application of AVOIR more broadly without having to hand derive rules for new metrics. Specifically, Verifair assumes that the data distribution is known and can be sampled from. To deal with unknown distributions, Verifair requires an additional fitting step of a density estimation model. The confidence set framework and inference rules in AVOIR enable AVOIR-VF to sidestep this assumption making AVOIR-VF more efficient than the original Verifair approach. We further demonstrate that AVOIR-OB provably converges in at most as many iterations as AVOIR-VF (empirically we see it to be 10% fewer on some datasets).
>
> > W6: What exactly is being interactively tuned?
>
> We have clarified the language on this aspect to reference the ability to diagnose issues with fairness specifications through subexpression confidence sets (as shown in the Adult income study (3.2). Note that the use of the Adaptive Hoeffding inequality is only possible with an IID test distribution. Thus, the train and test distributions may be unknown and have a mismatch which makes our work more general than Bastani et al. Non stationary data is beyond the scope of our work, and we have added this is the future work (Section 4).
>
> > Nits:
>
> We thank the review for helping contextualize our work better w.r.t Nissenbaum’s work (added reference); we see our work as a mechanism of helping both developers and regulators of machine learning systems better account for claims of mathematical guarantees. The COMPAS case study (Appendix G.3) is one such instance provided in our paper.

---

### Official Review · Reviewer_Gatp · 2022-11-05

**Confidence:** 2
**Correctness:** 3
**Technical Novelty And Significance:** 3
**Empirical Novelty And Significance:** 2
**Recommendation:** 3

**Clarity, Quality, Novelty And Reproducibility:**

The proofs support theoretical results, and sufficient experiment/implementation details are provided for reproducibility.

The idea of improving using optimization for improving bounds for fairness auditing is novel, but problem setup and other ideas have been explored in prior work.


**Strength And Weaknesses:**

Strengths:
- The observation that different (sub-)expressions usually have different power (N) is interesting, and the empirical benefits from improving concentration bounds are significant.
- Authors create an automated system leveraging existing tools, which can be helpful in practical settings.

Weaknesses:
- The paper is unclear at many points and hard to parse:
	- The definition and difference b/w different terms, such as spec, expression, claim, etc., needs to be clarified. It would be good to explain all these with concrete examples in one place, in a preliminary or background section.
	- The example in sec 3.2 is unclear. How does X: ..., Y: ... translate to  X \pm Y.
	- Some figures do not have legends and labels, making the results hard to interpret.
- Corollary 4.2 is unclear. What is c? Moreover, it appears that the algorithm would not stop/cannot detect if the specification is "almost surely" violated in a limited number of samples, i.e., if Alg 1, line 18 has no solution, it would keep running. So if the expectation converges to some other value than what is specified, i.e., fairness constraints are not satisfied, it won't terminate or throw an error.
- No human/practitioner study is done to assess the usability of the visualization & auditing tool.

Questions & other minor errors:
- Fig 3b, 4a, 4b : What are x and y axis, legends in 4a 4b?
- The fairness specification in sec 4.1 has a typo?
- What does the heading adaptive optimization mean in Table 1?
- Why do authors use $\delta_i=\delta_j$, when prior works have used $\Delta/n (as also mentioned in the paper)$? How do the results change with $\Delta/n$

Typos:
 - stochaistic -> stochastic (Sec 1, second last para)
- "choose an a suitable" -> choose a suitable (sec 1, last para)
- Sec 3.1, first para "We then infer a symbolic optimization problem is inferred corresponding"


**Summary Of The Paper:**

This paper improves on previous methods for automated fairness auditing. Their main contributions are improving the concentration bounds, extending the language to specify fairness metrics, and creating a visualization tool.

- For complex fairness metrics/expressions, the authors propose to distribute error rates unequally by solving an optimization problem. These can reduce the number of samples required to verify the conditions.
- Building on prior work, the authors provide an automated way to specify and track the fairness metrics.
- They also create a visualization tool for practitioners to visualize which sub-expressions are violated.

**Summary Of The Review:**

These works heavily build on prior work, and the novelty is unclear. The main novelty in this work is improving the number of samples required to audit fairness,  which seems to be limited. Moreover, the paper is hard to parse and unclear in many places. Thus, I suggest rejection.

---

> ### Author Response · Authors · 2022-11-16
> **Response to Reviewer Gatp**
>
> We thank the reviewer for their comprehensive review. Individual responses below:
> Responses to weaknesses
>
> **On clarity:**
> >The definition and difference b/w different terms, such as spec, expression, claim, etc., needs to be clarified. It would be good to explain all these with concrete examples in one place, in a preliminary or background section.
>
> We added a preliminaries section (1.1) to the introduction to introduce the terminology. We also expand the language specification section (2.1) and provide examples of supported metrics in Appendix H.
>
> >The example in sec 3.2 is unclear. How does X: ..., Y: ... translate to X \pm Y.
>
> The assertions on $X$ and $Y$ are assumed as preconditions to derive bounds on $X \pm Y$. This is also true of all other bound inference rules. Bounds on $X$ and $Y$ are derived from concentration inequalities. We update the language in Section 2.2 to reflect this.
>
> >Some figures do not have legends and labels, making the results hard to interpret.
>
> We have updated all figures with legends and labels
>
> **Corollary 4.2 and convergence**
> >Corollary 4.2 is unclear. What is c? Moreover, it appears that the algorithm would not stop/cannot detect if the specification is "almost surely" violated in a limited number of samples, i.e., if Alg 1, line 18 has no solution, it would keep running. So if the expectation converges to some other value than what is specified, i.e., fairness constraints are not satisfied, it won't terminate or throw an error.
>
> $c$ corresponds to a constant threshold involved in any specification. We clarified this in the updated version.
>
> **Convergence**: The bounds are computed with respect to the empirical value of expectations. Therefore if the specification is violated, the optimization problem solved will correspond to a guarantee that the specification is violated with probability $geq 1 - \Delta$. The COMPAS case study in G.3 has an example of termination with specification violation.
>
> > No human/practitioner study is done to assess the usability of the visualization & auditing tool.
>
> We would like to clarify that the visualization is mainly used as a diagnostic tool for specifications, and have also added this in the overall response and contributions section (1.3) of the revised paper. We have also demonstrated this in the Adult Income data case study (Section 3.2).
>
> **Questions & minor concerns**:
> * We have updated the figures with legends and labels.
> * The specification in 4.1 was copied from the python implementation and had a stray \ for a multiline specification. We thank the reviewer for helping us correct this.
> * $\delta_i = \delta_j,, \sum \delta_i = \Delta \implies \delta_i = \Delta/n$.
>
> **On novelty**
> We have clarified how our work is distinct from prior work in the revised contributions section (1.3) and have also highlighted our contributions in the [overall response](https://openreview.net/forum?id=Gp91Et4LeRf&noteId=hx4RuvvITrh) to all reviewers. To reiterate, the key differentiators of AVOIR are
> * AVOIR relies on adaptive optimization via a unique confidence set framework
> * More efficient Bound Propagation
> * Inference Engine.

---

### Author Response · Authors · 2022-11-16
**General Response**

On reading the reviews, we recognize that we may have oversimplified the nuances of our contributions through the use of the Table with checkmark. To remedy these issues, we have removed this table and instead added a comprehensive related work section. We also clarify our contributions vis-à-vis related work below:

**Note**: FP refers to Fairness-aware Programming (Albarghouthi & Vinitsky, 2019) and Verifair refers to (Bastani et al 2019).

We also want to clarify the distinction between Verifair and AVOIR-VF. Verifair assumes that the data distribution is known and can be sampled from, and therefore requires an additional fitting step for density estimation. The confidence set framework and inference rules in AVOIR enable AVOIR-VF to sidestep this assumption making AVOIR-VF more efficient than the original Verifair approach.

AVOIR-OB incorporates all the optimizations that we develop in our paper, and represents an improvement over AVOIR-VF and Verifair.

* **AVOIR relies on adaptive optimization via a unique confidence set framework**
    * We build up AVOIR in the framework of confidence sets. This is a unique theoretical contribution. We prove that the optimization problem can be solved at each step if we follow a specific strategy of assigning uncertainty that is enabled by the confidence sets framework.
    * **AVOIR adaptively splits uncertainty across expressions unlike FP and Verifair**
        * FP has no particular provision about splitting uncertainty across terms. They provide an example with equal splits across two terms.
        * Verifair splits uncertainty equally across all terms ($\Delta/n$).
    * **Verifair assumes that the data distribution is known and can be sampled from.**
        * Specifically, in section 3.1 in the Bastani et al, Verifair assumes that data is generated from a  population distribution over a known population model, and that their algorithm can sample from conditional distributions to estimate conditional expectations.
        * Applying Verifair to real-world data requires fitting a density estimation model over a population, using techniques such as GANs or VAEs before deriving any bounds for fairness specifications (as stated by Section 7 in the Verifair paper). We do not require this additional step.
        * Note that the confidence sets framework allows us to circumvent this distributional assumption and actually leverage Verifair bound propagation rules (which is dubbed AVOIR-VF in our paper).
* **Bound Propagation Rules in AVOIR are efficient**
    * **The error propagation rules in FP are relatively inefficient.**
        * Specifically, FP rules for binary operations (+, -, *, /) use actual values with every possible combination of operators  to compute the overall web.wha
        * This leads to the addition of at least 4 constraints per binary operator subexpression. All rules in AVOIR add at most one constraint per rule.
    * **FP also uses an inefficient concentration bound.**
        * FP uses the trivial version of the asymptotic Hoeffding bound, and as can be seen from Fig 2(a) in our paper,these are suboptimal.
    * **Verifair lacks constraint propagation rules for bounds**
        * In contrast with Verifair, we introduce additional constraint propagation rules into the bounds propagation sections. These rules enable AVOIR to optimize the distribution of uncertainty across different subexpressions and converge to a guarantee in fewer steps.
* **AVOIR provides an Inference Engine.**
    *  The implementation provided by the Verifair authors only provides support for specific (fixed) metrics. We provide an automated inference engine that can be used to run Verifair propagation rules that support a much wider range of metrics automatically. This model is dubbed AVOIR-VF in our paper.
    * This is also evident in the examples chosen by Bastani et al and FP - they only demonstrate their frameworks on at most two elementary subexpressions (where the inference is easier to analytically derive). We show examples with more terms as our engine can automatically derive and optimize; we do not need to hardcode the rules.
* **AVOIR supports interactive diagnosis of fairness specification violations via visual cues**
    * As demonstrated in the Adult Income case study, the initial specification for gender fairness cannot be guaranteed with the given number of samples. However, investigating the subexpressions involved reveals that one of the subexpressions converges at a slower rate. Using this observation, the corresponding subexpression in the specification can be updated, and the updated specification (with a weaker guarantee) can be reached with the available data samples.

---

### Decision · Program_Chairs · 2023-01-20

**Decision:**

Reject

**Justification For Why Not Higher Score:**

The reviewers highlighted an extensive list of concerns, most prominently in terms of presentation, contextualization with related work, and novelty and were not in support of accepting the paper.

**Justification For Why Not Lower Score:**

N/A

**Metareview: Summary, Strengths And Weaknesses:**

The authors introduce a fairness audit framework that allows the generation of fairness probabilistic guarantees for any models and fairness metrics. None of the reviewers were in support of accepting this paper. They highlighted an extensive list of concerns, most prominently in terms of presentation, contextualization with related work, and novelty. As a result, I am afraid I cannot recommend acceptance.